# Disruption of the CRF$_1$ receptor eliminates morphine-induced sociability deficits and firing of oxytocinergic neurons in male mice

Alessandro Piccin[1,2], Anne-Emilie Allain[1,2], Jérôme M Baufreton[3,4], Sandrine S Bertrand[1,2], Angelo Contarino[1,2,5]*

[1]Université de Bordeaux, INCIA, Bordeaux, France; [2]CNRS, INCIA, Bordeaux, France; [3]Université de Bordeaux, IMN, Bordeaux, France; [4]CNRS, IMN, Bordeaux, France; [5]INSERM, T3S, UMR-S 1124, Université Paris Cité, Paris, France

## eLife Assessment

The revised report provides **valuable** findings for the field, suggesting a relationship between CRF1 receptors, sociability deficits in morphine-treated male mice yet not females, and a potential mechanism involving oxytocin neurons in the paraventricular nucleus of the hypothalamus. Generally, the strength of evidence is **solid** in terms of the methods, data, and analyses. This work will be of interest to those interested in social behavior and addiction.

*For correspondence:
angelo.contarino@u-bordeaux.fr

Competing interest: The authors declare that no competing interests exist.

**Abstract** Substance-induced social behavior deficits dramatically worsen the clinical outcome of substance use disorders; yet, the underlying mechanisms remain poorly understood. Herein, we investigated the role for the corticotropin-releasing factor receptor 1 (CRF$_1$) in the acute sociability deficits induced by morphine and the related activity of oxytocin (OXY)- and arginine-vasopressin (AVP)-expressing neurons of the paraventricular nucleus of the hypothalamus (PVN). For this purpose, we used both the CRF$_1$ receptor-preferring antagonist compound antalarmin and the genetic mouse model of CRF$_1$ receptor-deficiency. Antalarmin completely abolished sociability deficits induced by morphine in male, but not in female, C57BL/6J mice. Accordingly, genetic CRF$_1$ receptor-deficiency eliminated morphine-induced sociability deficits in male mice. Ex vivo electrophysiology studies showed that antalarmin also eliminated morphine-induced firing of PVN neurons in male, but not in female, C57BL/6J mice. Likewise, genetic CRF$_1$ receptor-deficiency reduced morphine-induced firing of PVN neurons in a CRF$_1$ gene expression-dependent manner. The electrophysiology results consistently mirrored the behavioral results, indicating a link between morphine-induced PVN activity and sociability deficits. Interestingly, in male mice antalarmin abolished morphine-induced firing in neurons co-expressing OXY and AVP, but not in neurons expressing only AVP. In contrast, in female mice antalarmin did not affect morphine-induced firing of neurons co-expressing OXY and AVP or only OXY, indicating a selective sex-specific role for the CRF$_1$ receptor in opiate-induced PVN OXY activity. The present findings demonstrate a major, sex-linked, role for the CRF$_1$ receptor in sociability deficits and related brain alterations induced by morphine, suggesting new therapeutic strategy for opiate use disorders.

## Introduction

Opiate substances often induce severe social behavior deficits, such as poor sociability, social isolation, and elevated aggressiveness (*Babor et al., 1976*; *Gerra et al., 2004*). Notably, opiate-induced social behavior deficits dramatically contribute to addictive-like substance consumption, favoring the development and maintenance of opiate use disorders (OUD) (*APA, 2013*; *Pomrenze et al., 2022*). Thus, it has been hypothesized that treatments increasing positive peer relationships might considerably reduce substance seeking and taking and ameliorate the clinical outcome of substance-dependent patients (*Heilig et al., 2016*; *Venniro et al., 2018*). However, the development of novel, effective, therapy heavily relies on a better understanding of the brain mechanisms underlying the harmful effects of substances of abuse; yet, to date the neurobiological substrates of substance-induced social behavior deficits remain largely unknown.

The corticotropin-releasing factor (CRF) system is a main orchestrator of behavioral and neuroendocrine responses to stress (*Dedic et al., 2018*; *Koob, 2008*). The CRF system might also underlie the behavioral and brain effects of substances of abuse (*Koob, 2008*). CRF signaling is mediated by two types of receptors, named $CRF_1$ and $CRF_2$ (*Hauger et al., 2003*). Relatively recent studies shed some light on the role for each of the two known CRF receptor types in social behavior deficits induced by repeated administration of and withdrawal from substances of abuse. For instance, gene knockout (KO) of the $CRF_2$ receptor reduced sociability deficits and vulnerability to stress associated with long-term cocaine withdrawal in male mice (*Morisot et al., 2018*). Moreover, genetic inactivation of the $CRF_1$ receptor ($CRF_1$ KO) decreased morphine withdrawal-induced sociability deficits in female mice and hostility-driven interest for a same-sex conspecific in male mice (*Piccin and Contarino, 2022a*). The CRF system may also interact with other brain systems implicated in social behavior. In particular, extensive literature points out to the two closely related neuropeptides oxytocin (OXY) and arginine-vasopressin (AVP) as main substrates of social interaction, parenting behavior and intermale aggressiveness (*Jurek and Neumann, 2018*). For instance, chemogenetic activation or inhibition of OXY-expressing neurons within the paraventricular nucleus of the hypothalamus (PVN), respectively, increased or decreased social approach in male mice (*Resendez et al., 2020*). Accordingly, social stimuli increased the activity of PVN OXY-expressing neurons, as assessed by in vivo two-photon calcium imaging (*Resendez et al., 2020*). Moreover, coordinated responses of PVN parvocellular and magnocellular OXY-expressing neurons to somatosensory stimuli mediated social interaction in female rats (*Tang et al., 2020*). In contrast, male *Shank3b* knockout mice, that is, a mouse model of autistic-like behavior, showed a marked reduction in PVN OXY-expressing neurons and decreased social approach (*Peça et al., 2011*; *Resendez et al., 2020*). Furthermore, targeted chemogenetic silencing of PVN OXY neurons in male rats impaired short- and long-term social recognition memory (*Thirtamara Rajamani et al., 2024*). Likewise OXY and AVP, CRF is largely expressed in the PVN (*Jiang et al., 2018*; *Sawchenko et al., 1993*). Interestingly, whole-cell patch-clamp studies showed that CRF- and OXY-expressing neurons are highly intermingled within the PVN, suggesting local cell-to-cell interactions (*Jamieson et al., 2017*). CRF might also modulate hypothalamic OXY and AVP responses to substances of abuse. For instance, long-term cocaine-withdrawn male $CRF_2$ KO mice showed neither the stress-induced sociability deficits nor the related increased expression of OXY or AVP in the supraoptic nucleus of the hypothalamus (SON) or the PVN (*Morisot et al., 2018*). Thus, CRF, OXY, and AVP systems may be potential targets of effective therapy for diseases characterized by dysfunctional social behavior, including substance use disorders. However, to date very little is known about their implication in social behavior deficits induced by substances of abuse.

Thus, herein we investigated the role for the CRF/$CRF_1$ receptor pathway in the acute social behavior deficits following opiate administration. In particular, using the three-chamber task for sociability in mice, the role for the $CRF_1$ receptor in sociability deficits induced by morphine was assessed by both pharmacological (i.e., the $CRF_1$ receptor-preferring antagonist antalarmin) and genetic (i.e., the $CRF_1$ receptor-deficient mouse model) approaches (*Moy et al., 2004*; *Smith et al., 1998*; *Webster et al., 1996*). Moreover, to understand CRF role in brain OXY and AVP responses to morphine, ex vivo electrophysiology studies assessed the effect of antalarmin and genetic $CRF_1$ receptor-deficiency upon morphine-induced firing of PVN OXY- and/or AVP-immunoreactive neurons. Notably, to fully adhere to the Sex as a Biological Variable (SABV) initiative and given the well-established influence of sex upon the addictive-like properties of substances of abuse, herein male and female mice were used throughout (*Becker and Koob, 2016*; *Clayton, 2018*).

## Results

### Pharmacological CRF$_1$ receptor antagonism eliminates morphine-induced sociability deficits in male, but not in female, mice

The acute effects of morphine upon social behavior were investigated using the three-chamber test for sociability in mice, as previously reported (*Piccin et al., 2022b*). During the habituation phase of the test, male C57BL/6J mice spent similar time in the regions of interest (ROIs, side half-chambers) of the apparatus (*Figure 1A, B* and *Supplementary file 1d*). Analysis of the sociability phase revealed a *pretreatment × treatment × repeated measures* interaction effect (*Supplementary file 1d*). Unlike saline-treated mice, vehicle/morphine-treated mice spent similar time in the ROIs containing the unfamiliar conspecific or the object (p = 0.823), indicating sociability deficits (*Figure 1C*). In contrast, antalarmin/morphine-treated mice spent more time with the unfamiliar conspecific than with the object (p < 0.005), indicating unaltered sociability (*Figure 1C*). Accordingly, analysis of sociability ratio revealed a *pretreatment* effect ($F_{1,32}$ = 11.598, p < 0.005), a *treatment* effect ($F_{1,32}$ = 4.713, p < 0.05) and a *pretreatment × treatment* interaction effect ($F_{1,32}$ = 8.718, p < 0.01). Vehicle/morphine-treated mice showed lower sociability ratio than vehicle/saline-treated mice (p < 0.005, *Figure 1D*). In contrast, antalarmin/morphine-treated mice did not differ from saline-treated mice (p = 0.661) and showed higher sociability ratio than vehicle/morphine-treated mice (p < 0.005, *Figure 1D*). During the habituation phase, morphine-treated female C57BL/6J mice spent less time in the ROIs of the apparatus than saline-treated mice (p < 0.0005), independently of vehicle or antalarmin pretreatment (*Figure 1E* and *Supplementary file 1d*). Moreover, analysis of the sociability phase revealed a *treatment × repeated measures* interaction effect but no *pretreatment × treatment × repeated measures* interaction effect (*Supplementary file 1d*). Indeed, unlike saline-treated mice, morphine-treated mice spent similar time in the ROIs containing the unfamiliar conspecific or the object (p = 0.259), independently of vehicle or antalarmin pretreatment (*Figure 1F*). Accordingly, analysis of sociability ratio revealed no *pretreatment* effect ($F_{1,21}$ = 0.035, p = 0.852), a *treatment* effect ($F_{1,21}$ = 8.698, p < 0.01) but no *pretreatment × treatment* interaction effect ($F_{1,21}$ = 0.018, p = 0.894). Morphine-treated mice showed lower sociability ratio than saline-treated mice (p < 0.05), independently of vehicle or antalarmin pretreatment (*Figure 1G*). Sociability ratio was also examined by a three-way ANOVA with sex (males vs. females), pretreatment (vehicle vs. antalarmin) and treatment (saline vs. morphine) as between-subjects factors. The latter analysis revealed an almost significant *sex × pretreatment × treatment* interaction effect ($F_{1,53}$ = 3.287, p = 0.075), which could not allow for post hoc individual group comparisons. Nevertheless, post hoc tests revealed that male mice treated with antalarmin/morphine showed higher sociability ratio than female mice treated with antalarmin/morphine (p < 0.05). During the three-chamber test, morphine-treated male, but not female, mice traveled more distance than saline-treated mice (p < 0.05), independently of vehicle or antalarmin pretreatment (*Supplementary file 1e* and *Figure 1—figure supplement 1A, B*). Also, overall mice traveled more distance during the habituation than during the sociability phase (p < 0.05), indicating familiarization with the test apparatus (*Supplementary file 1e* and *Figure 1—figure supplement 1A, B*). Thus, the present results indicate a sex-linked role for the CRF$_1$ receptor in social behavior deficits induced by morphine. Moreover, morphine effects upon sociability seemed unrelated to locomotor activity.

### Pharmacological CRF$_1$ receptor antagonism eliminates morphine-induced firing of PVN neurons in male, but not in female, mice

To investigate the neural substrates of CRF$_1$ receptor-mediated sociability deficits induced by morphine, electrophysiology studies examined firing frequency of PVN neurons (*Figure 2A*). In male C57BL/6J mice, analysis of firing frequency of all of the recorded cells (n = 110) revealed a *pretreatment* effect ($F_{1,106}$ = 7.894, p < 0.01), a *treatment* effect ($F_{1,106}$ = 13.350, p < 0.0005) and a *pretreatment × treatment* interaction effect ($F_{1,106}$ = 4.208, p < 0.05). Vehicle/morphine-treated mice showed higher firing frequency than vehicle/saline-treated mice (p < 0.0005, *Figure 2B*). In contrast, antalarmin/morphine-treated mice did not differ from saline-treated mice (p = 0.552) and showed lower firing frequency than vehicle/morphine-treated mice (p < 0.005, *Figure 2B*). On the other hand, analysis of firing frequency of all of the recorded cells (n = 93) in female C57BL/6J mice revealed no *pretreatment* effect ($F_{1,89}$ = 0.049, p = 0.826), a *treatment* effect ($F_{1,89}$ = 20.476, p < 0.0001) but no *pretreatment × treatment* interaction effect ($F_{1,89}$ = 1.045, p = 0.310). Morphine similarly increased firing frequency in

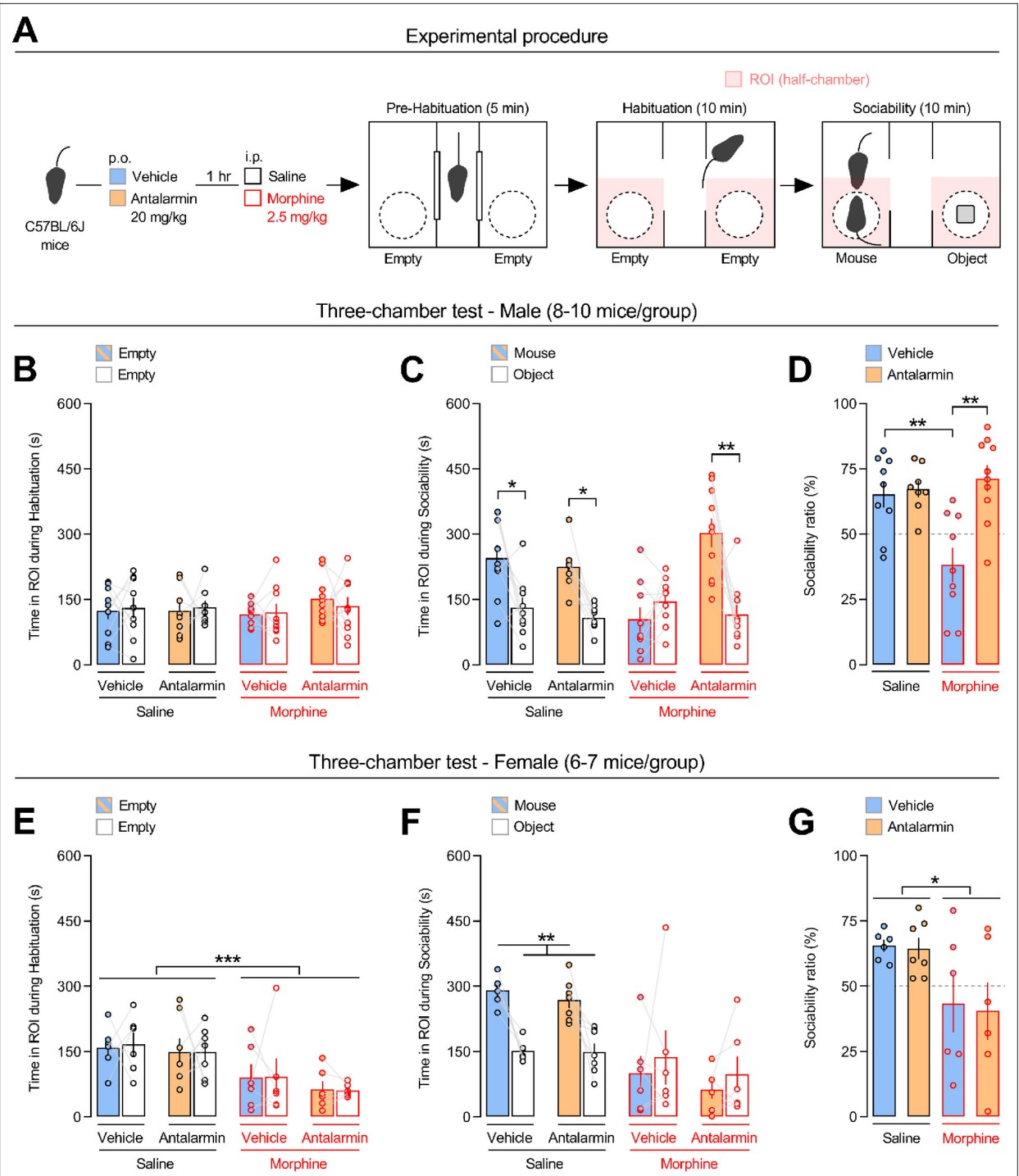

**Figure 1.** Pharmacological antagonism of the CRF$_1$ receptor eliminates morphine-induced sociability deficits in male, but not in female, mice. (**A**) Experimental procedure. Male and female C57BL/6J mice were injected per os (p.o.) with either vehicle or the CRF$_1$ receptor-preferring antagonist antalarmin (20 mg/kg). One hour later, they were injected intraperitoneally (i.p.) with either saline or morphine (2.5 mg/kg) and tested in the three-chamber task for sociability. Time (s) spent in the regions of interest (ROIs, side half-chambers) of the three-chamber apparatus by male (**B, C**) and female (**E, F**) mice during the (**B, E**) habituation or the (**C, F**) sociability phase of the test. During the habituation phase, the ROIs contained empty wire cages; during the sociability phase, the wire cages contained an unfamiliar same-sex mouse or an object (**A**). Sociability ratio (%) displayed by (**D**) male and (**G**) female mice. The number of animals within each experimental group is reported in *Supplementary file 1a*. Values represent mean ± SEM. *p < 0.05, **p < 0.005, ***p < 0.0005.

The online version of this article includes the following figure supplement(s) for figure 1:

**Figure supplement 1.** Locomotor activity of C57BL/6J mice during the three-chamber test with morphine.

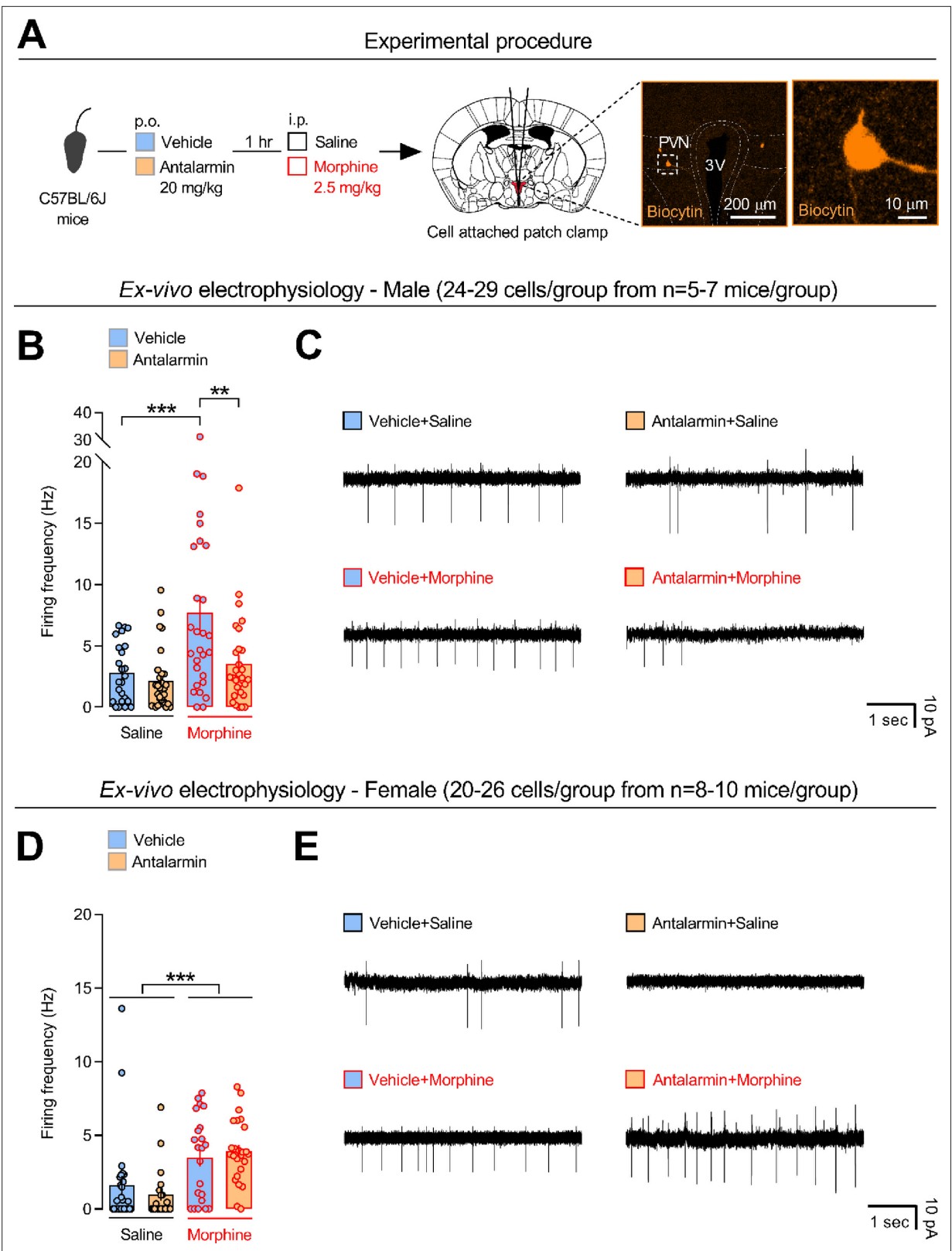

**Figure 2.** Pharmacological antagonism of the CRF$_1$ receptor eliminates neuronal firing induced by morphine in male, but not in female, mice. (**A**) Experimental procedure. Male and female C57BL/6J mice were injected per os (p.o.) with either vehicle or the CRF$_1$ receptor-preferring antagonist antalarmin (20 mg/kg). One hour later, they were injected intraperitoneally (i.p.) with either saline or morphine (2.5 mg/kg). Ten minutes after, brains were removed and cell-attached patch-clamp recordings of paraventricular nucleus of the hypothalamus (PVN) neurons performed from brain slices.

*Figure 2 continued on next page*

*Figure 2 continued*

Scale bars: 200 and 10 µm. Firing frequency (Hz) of PVN neurons displayed by (**B**) male and (**D**) female mice treated with either vehicle or antalarmin followed by either saline or morphine. Images showing electrophysiological recordings from PVN neurons of the four (**C**) male and the four (**E**) female experimental groups. The number of total patched and recorded cells within each experimental group is reported in *Supplementary file 1c*. Values represent mean ± SEM. **p < 0.005, ***p < 0.0005.

vehicle- or antalarmin-pretreated mice, as compared to saline-treated mice (p < 0.0005, *Figure 2D*). Firing frequency of all of the recorded cells was also examined by a three-way ANOVA with sex (males vs. females), pretreatment (vehicle vs. antalarmin), and treatment (saline vs. morphine) as between-subjects factors. The latter analysis revealed a *sex × pretreatment × treatment* interaction effect ($F_{1,195}$ = 4.765, p < 0.05). Post hoc individual group comparisons revealed that male mice treated with vehicle/morphine showed higher firing frequency than all other male and female groups (p < 0.0005). Moreover, male mice treated with antalarmin/morphine showed lower firing frequency than male mice treated with vehicle/morphine (p < 0.0005). In contrast, female mice treated with antalarmin/morphine did not differ from female mice treated with vehicle/morphine (p = 0.914). These results indicate a critical role for the $CRF_1$ receptor in PVN neuronal activity induced by morphine in male, but not in female, mice. Notably, the sex-dependent effects of antalarmin upon neuronal firing closely mimicked the social behavior results, indicating a link between PVN activity and sociability deficits induced by morphine.

### Genetic inactivation of the $CRF_1$ receptor eliminates morphine-induced sociability deficits

The specific role for the $CRF_1$ receptor in morphine-induced sociability deficits was further investigated using the genetic mouse model of $CRF_1$ receptor-deficiency. We first tested n = 3 $CRF_1$ wild-type (WT) and n = 3 $CRF_1$ heterozygote (HET) male mice using the same morphine dose (2.5 mg/kg) employed in the C57BL/6J mice. However, during the whole 10-min habituation phase of the three-chamber test, all of the six morphine-treated animals remained in the central chamber of the apparatus, suggesting that the morphine dose used was relatively high. Thus, we decided to use a substantially lower morphine dose, that is, 0.625 mg/kg (*Figure 3A*). During the habituation phase, morphine (0.625 mg/kg) reduced the time spent in both ROIs of the three-chamber apparatus in $CRF_1$ HET (p < 0.05 vs. saline-treated $CRF_1$ HET mice), but not in $CRF_1$ WT or $CRF_1$ KO, male mice (*Figure 3B* and *Supplementary file 1f*). Analysis of the sociability phase revealed a *genotype × treatment × repeated measures* interaction effect (*Supplementary file 1f*). Unlike saline-treated mice, morphine-treated $CRF_1$ WT and $CRF_1$ HET mice spent similar time in the ROIs containing the unfamiliar conspecific or the object (p = 0.873), indicating sociability deficits (*Figure 3C*). In contrast, morphine-treated $CRF_1$ KO mice spent more time with the conspecific than with the object (p < 0.005), indicating unaltered sociability (*Figure 3C*). Accordingly, analysis of sociability ratio revealed no *genotype* effect ($F_{2,58}$ = 2.641, p = 0.080), a *treatment* effect ($F_{1,58}$ = 7.478, p < 0.01), and a *genotype × treatment* interaction effect ($F_{2,58}$ = 4.994, p < 0.01). Morphine-treated $CRF_1$ WT mice showed lower sociability ratio than saline-treated $CRF_1$ WT mice (p < 0.05, *Figure 3D*). In contrast, morphine-treated $CRF_1$ KO mice did not differ from saline-treated mice (p = 0.819) and showed higher sociability ratio than morphine-treated $CRF_1$ WT and $CRF_1$ HET mice (p < 0.05, *Figure 3D*). Interestingly, unlike $CRF_1$ WT and $CRF_1$ KO mice, morphine-treated $CRF_1$ HET mice almost differed from saline-treated $CRF_1$ HET mice (p = 0.065), suggesting a gene expression-dependent effect of $CRF_1$ receptor-deficiency (*Figure 3D*). During the three-chamber test, overall morphine did not affect distance traveled (*Supplementary file 1f*). Moreover, saline-treated mice and morphine-treated $CRF_1$ WT, but not $CRF_1$ HET or $CRF_1$ KO, mice traveled more distance during the habituation than during the sociability phase (p < 0.05, *Figure 3—figure supplement 1* and *Supplementary file 1f*), further indicating dissociation between the locomotor and the sociability effects of morphine. Thus, the similar results obtained with $CRF_1$ KO and antalarmin-treated C57BL/6J male mice strengthened the notion of a key role for the $CRF_1$ receptor in sociability deficits induced by morphine.

We then assessed the effect of morphine (0.625 mg/kg) in female $CRF_1$ receptor-deficient mice. However, as shown in *Supplementary file 1g*, during the habituation phase of the three-chamber test only 2/8 $CRF_1$ WT mice treated with saline and 2/8 $CRF_1$ WT mice treated with morphine visited both side chambers of the apparatus. Also, despite all saline-treated $CRF_1$ HET mice (n = 4) visited both

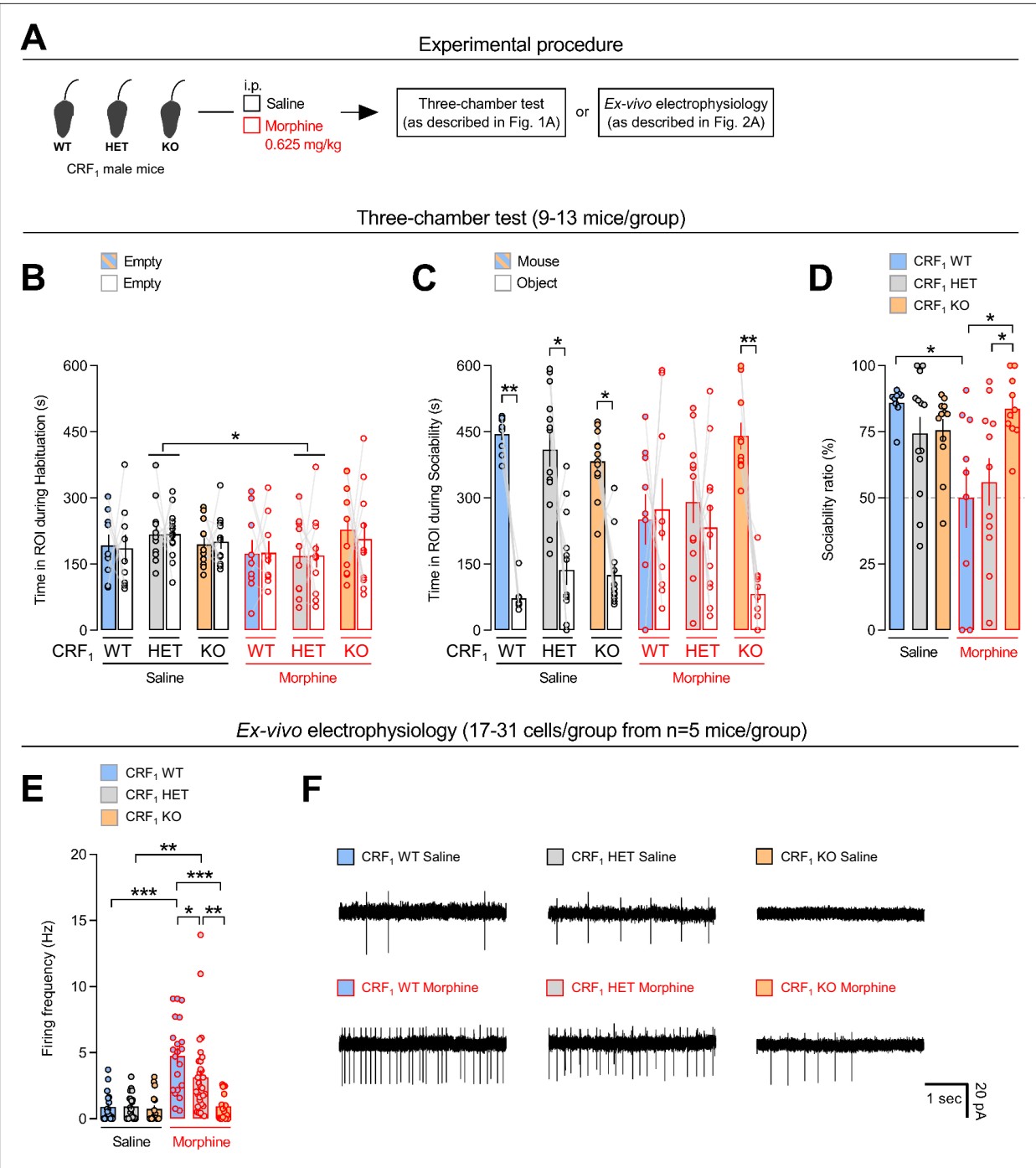

**Figure 3.** Genetic inactivation of the CRF$_1$ receptor eliminates morphine-induced sociability deficits and neuronal firing. (**A**) Experimental procedure. Male CRF$_1$ WT, CRF$_1$ HET, and CRF$_1$ KO mice were injected intraperitoneally (i.p.) with either saline or morphine (0.625 mg/kg) and tested in the three-chamber task for sociability. Additional groups of male CRF$_1$ WT, CRF$_1$ HET, and CRF$_1$ KO mice were injected with either saline or morphine (0.625 mg/kg) and cell-attached patch-clamp recordings of paraventricular nucleus of the hypothalamus (PVN) neurons performed from brain slices. Time (s) spent in the regions of interest (ROIs, side half-chambers) of the three-chamber apparatus (see *Figure 1A*) during the (**B**) habituation and the (**C**) sociability phase of the test, (**D**) sociability ratio (%) and (**E**) firing frequency (Hz) of PVN neurons by saline- or morphine-treated CRF$_1$ WT, CRF$_1$ HET and CRF$_1$ KO mice. (**F**) Images showing electrophysiological recordings from PVN neurons of the six experimental groups. The number of animals and the number of patched and recorded cells within each experimental group are reported in *Supplementary file 1b*. Values represent mean ± SEM. *p < 0.05, **p < 0.005, ***p < 0.0005.

The online version of this article includes the following figure supplement(s) for figure 3:

**Figure supplement 1.** Locomotor activity of CRF$_1$ receptor-deficient mice during the three-chamber test with morphine.

side chambers of the apparatus, this occurred only in 3/8 CRF$_1$ HET mice treated with morphine. Thus, we could not obtain a reliable amount of data using a reasonable number of female mice, at least under our experimental conditions and with the 0.625 mg/kg morphine dose.

## Genetic CRF$_1$ receptor-deficiency eliminates morphine-induced firing of PVN neurons

Analysis of firing frequency of PVN neurons in male CRF$_1$ receptor-deficient mice revealed a *genotype* effect ($F_{2,119}$ = 8.498, p < 0.0005), a *treatment* effect ($F_{1,119}$ = 31.816, p < 0.0001) and a *genotype × treatment* interaction effect ($F_{2,119}$ = 7.224, p < 0.005). Morphine (0.625 mg/kg) increased firing frequency in CRF$_1$ WT (p < 0.0005) and in CRF$_1$ HET (p < 0.005), but not in CRF$_1$ KO (p = 0.987), mice, as compared to same-genotype saline-treated mice (*Figure 3E*). Notably, morphine-treated CRF$_1$ HET mice showed lower or higher firing frequency than morphine-treated CRF$_1$ WT (p < 0.05) or CRF$_1$ KO (p < 0.005) mice, respectively, indicating a CRF$_1$ gene expression-dependent effect (*Figure 3E*). These results further support an essential role for the CRF$_1$ receptor in PVN neuronal firing induced by morphine. Moreover, the lack of morphine effects upon neuronal firing and sociability in CRF$_1$ KO mice indicates once more a link between PVN activity and social behavior.

## Pharmacological CRF$_1$ receptor antagonism eliminates morphine-induced firing of PVN OXY-expressing neurons in male, but not in female, mice

In male C57BL/6J mice, 40 cells expressed both OXY and AVP, 49 cells expressed AVP but not OXY, 6 cells expressed OXY but not AVP, and 15 cells expressed neither OXY nor AVP (*Figure 4B*). Vehicle/morphine-treated mice showed higher firing frequency of OXY/AVP-expressing neurons than vehicle/saline-treated mice (p < 0.0005; *Figure 4C* and *Supplementary file 1h*). In contrast, antalarmin/morphine-treated mice did not differ from saline-treated mice (p = 0.782) and showed lower firing frequency than vehicle/morphine-treated mice (p < 0.0005, *Figure 4C*). On the other hand, morphine increased firing frequency of neurons expressing AVP, but OXY, independently of vehicle or antalarmin pretreatment (p < 0.05; *Figure 4D* and *Supplementary file 1h*). In female C57BL/6J mice, 31 cells expressed both OXY and AVP, 38 cells expressed OXY but not AVP, 7 cells expressed AVP but not OXY, and 17 cells expressed neither OXY nor AVP (*Figure 4E*). Morphine increased firing frequency of neurons co-expressing OXY and AVP, independently of vehicle or antalarmin pretreatment (p < 0.005; *Figure 4F* and *Supplementary file 1h*). Similarly, morphine increased firing frequency of neurons expressing OXY, but not AVP, independently of vehicle or antalarmin pretreatment (p < 0.005; *Figure 4G* and *Supplementary file 1h*). These results indicate a sex-specific role for the CRF$_1$ receptor in morphine-induced firing of PVN OXY-expressing neurons, suggesting that CRF modulates brain OXY responses to opiate substances.

## Discussion

The present study demonstrates a major, sex-linked, role for the CRF$_1$ receptor in social behavior alterations induced by morphine. Indeed, male, but not female, mice treated with the CRF$_1$ receptor-preferring antagonist antalarmin did not show the sociability deficits induced by morphine. Accordingly, genetic inactivation of the CRF$_1$ receptor eliminated morphine-induced sociability deficits in male mice. Antalarmin also abolished morphine-induced firing of PVN neurons in male, but not in female, mice. Consistently, in male mice CRF$_1$ receptor-deficiency decreased morphine-induced firing of PVN neurons in a CRF$_1$ gene expression-dependent manner. Thus, the electrophysiology results reliably mirrored the behavioral results, suggesting a link between morphine-induced neuronal activity and sociability deficits. Furthermore, in male, but not in female, mice antalarmin eliminated morphine-induced firing in PVN neurons expressing OXY, suggesting sex-specific CRF–OXY interactions.

In agreement with our previous study (*Piccin et al., 2022b*), morphine consistently and similarly decreased sociability in male and female mice. Prior work reported sex-linked behavioral effects of opiate substances. For instance, female rats displayed greater motivation to take heroin and self-administered greater amounts of heroin or oxycodone than male rats (*Cicero et al., 2003*; *Fulenwider et al., 2020*; *George et al., 2021*). Moreover, female mice showed elevated heroin self-administration and increased sensitivity to the rewarding properties of morphine, as compared to male mice (*Piccin*

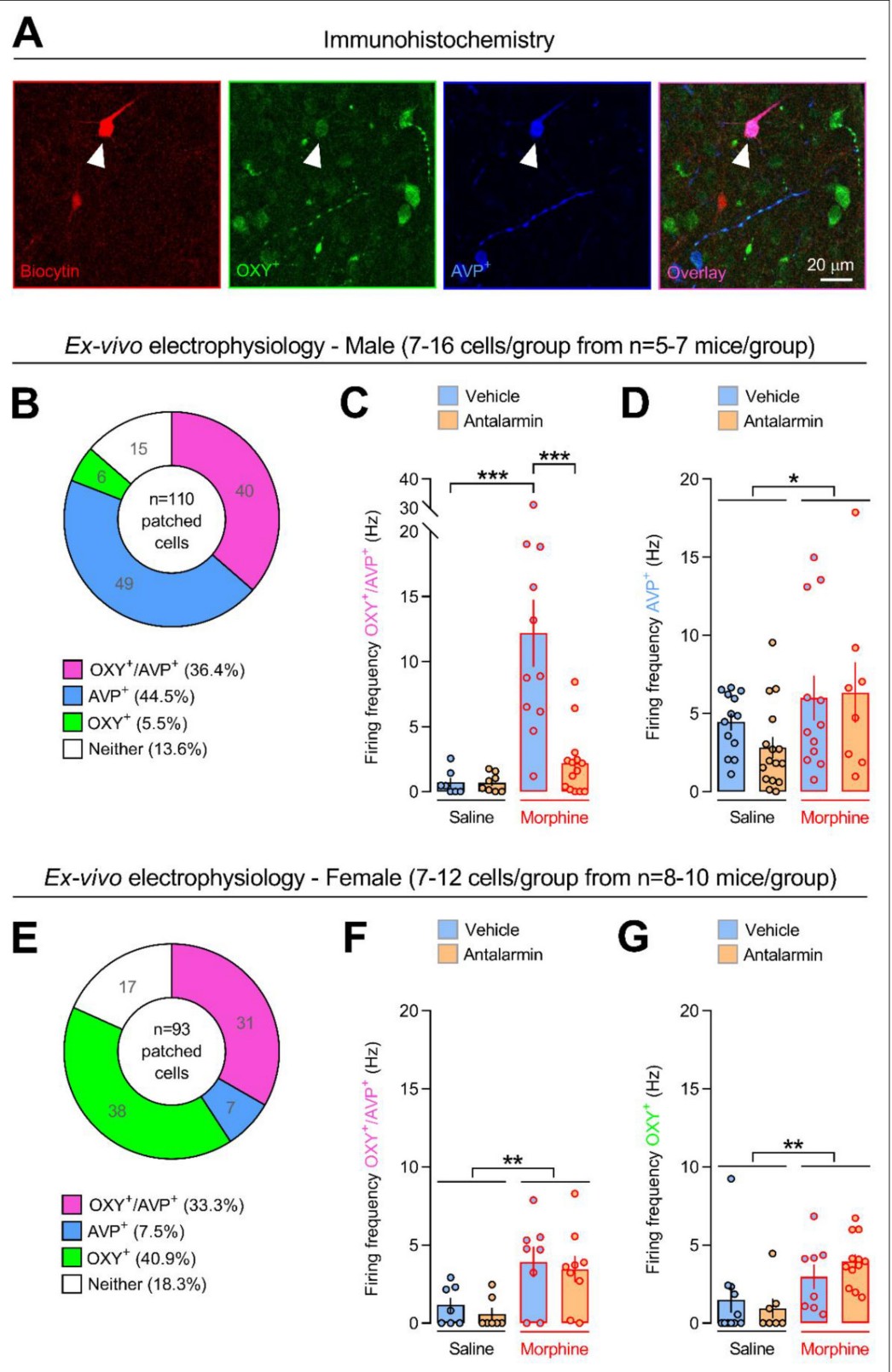

**Figure 4.** Pharmacological antagonism of the CRF$_1$ receptor eliminates morphine-induced firing of oxytocin-expressing neurons in male, but not in female, mice. (**A**) Immunohistochemical images of a paraventricular nucleus of the hypothalamus (PVN) neuron co-expressing oxytocin (OXY$^+$) and arginine-vasopressin (AVP$^+$). Scale bar: 20 μm. Number of patched and recorded PVN neurons expressing OXY and/or AVP or neither of the two

*Figure 4 continued on next page*

*Figure 4 continued*

neuropeptides in (**B**) male and (**E**) female C57BL/6J mice. Firing frequency (Hz) of PVN neurons expressing OXY and/or AVP in (**C, D**) male and (**F, G**) female C57BL/6J mice treated with either vehicle or antalarmin (20 mg/kg) followed by either saline or morphine (2.5 mg/kg), as shown in **Figure 2A**. The number of patched and recorded cells within each experimental group is reported in **Supplementary file 1c**. Values represent mean ± SEM. *p < 0.05, **p < 0.005, ***p < 0.0005.

*et al., 2022b*; *Towers et al., 2019*). Thus, unlike other behavioral effects of opiate substances, sex might have a marginal role in opiate-induced impairment of sociability. Nevertheless, herein $CRF_1$ receptor antagonism by antalarmin prevented morphine-induced sociability deficits in male, but not in female, mice. To date, very few studies have examined CRF role in social behavior effects of substances of abuse. For instance, genetic inactivation of the $CRF_2$ receptor reduced sociability deficits associated with long-term cocaine withdrawal in male mice (*Morisot et al., 2018*). Moreover, genetic $CRF_1$ receptor-deficiency decreased opiate withdrawal-induced sociability deficits in female mice, as assessed 1 week after cessation of repeated morphine administration (*Piccin and Contarino, 2022a*). The latter findings contrast with the present lack of effect of antalarmin in female mice. However, although it might be difficult to compare pharmacological and genetic studies, the possibility exists for a differential implication of the $CRF_1$ receptor in social behavior deficits induced by a single opiate administration or by withdrawal from repeated opiate administration. Nonetheless, the present findings might bear importance for opiate-related diseases since a single morphine administration may induce long-lasting behavioral and brain alterations relevant to substance use disorders (*Vanderschuren et al., 2001*). Thus, identifying the neural substrates underlying initial opiate effects might be critical to advance our quest toward the development of effective treatments for substance use disorders.

Genetically engineered mouse models might provide a level of molecular specificity that is rarely achieved by pharmacological tools. Thus, to specifically assess the role for the $CRF_1$ receptor in morphine-induced sociability deficits, herein $CRF_1$ receptor-deficient mice were also used (*Smith et al., 1998*). However, following preliminary experiments showing that male $CRF_1$ WT and $CRF_1$ HET mice treated with morphine (2.5 mg/kg) did not explore the three-chamber apparatus, a lower morphine dose was employed. Like in C57BL/6J mice, morphine (0.625 mg/kg) reliably impaired sociability in $CRF_1$ WT and $CRF_1$ HET male mice, indicating that the two morphine doses used herein were suitable to compare the effect of pharmacological and genetic disruption of the $CRF_1$ receptor. Some differences were though observed between $CRF_1$ receptor-deficient and C57BL/6J mice treated with saline. In particular, though we did not perform direct statistical comparisons, percentage of time spent with the unfamiliar conspecific seemed higher in saline-treated $CRF_1$ WT, $CRF_1$ HET, and $CRF_1$ KO male mice, as compared to saline-treated C57BL/6J male mice. It is difficult to understand the factors underlying the latter results. However, male mice bearing a mixed (B6x129PF2/J) genetic background also showed higher sociability levels than C57BL/6J male mice (*Moy et al., 2004*). Thus, it is possible that the mixed (C57BL/6Jx129S4/SvJae) genetic background of the $CRF_1$ receptor-deficient mice used herein contributed, at least in part, to increase social approach, as compared to inbred C57BL/6J mice. Nevertheless, despite the latter differences, $CRF_1$ receptor-deficiency ($CRF_1$ KO) completely eliminated the sociability deficits induced by morphine in male mice, further supporting the notion of an essential role for the $CRF_1$ receptor in opiate-induced disruption of social behavior. Unlike $CRF_1$ WT mice, $CRF_1$ KO mice showed sex-independent hypothalamus–pituitary–adrenal (HPA) axis deficits under basal and stressful conditions, as revealed by plasma adrenocorticotropic hormone (ACTH) and corticosterone assays (*Papaleo et al., 2007*; *Smith et al., 1998*; *Timpl et al., 1998*). Thus, it could be argued that the lack of morphine effects found in $CRF_1$ KO mice was due to HPA axis alterations. However, the present antalarmin results might, at least in part, rule out the latter hypothesis. Indeed, antalarmin is a non-peptide $CRF_1$ receptor-preferring antagonist that, upon systemic administration, readily crosses the blood–brain barrier and is behaviorally active (*Zorrilla and Koob, 2010*). Notably, antalarmin did affect neither basal nor stress-induced ACTH and corticosterone levels in male rats and mice (*Jutkiewicz et al., 2005*; *Pérez-Tejada et al., 2013*). Accordingly, the dose of antalarmin (20 mg/kg) used herein increased the somatic signs of morphine withdrawal without affecting plasma corticosterone (*Papaleo et al., 2007*). Thus, the present similar results obtained with antalarmin and

$CRF_1$ KO mice argue in favor of a marginal role for the HPA axis in $CRF_1$ receptor-mediated sociability deficits induced by morphine.

Throughout the present studies, locomotor activity during the three-chamber test did not seem to account for the $CRF_1$ receptor-mediated sociability deficits induced by morphine. For instance, antalarmin- and vehicle-treated male C57BL/6J mice showed similar locomotor but different sociability responses to morphine (*Figure 1D*, *Figure 1—figure supplement 1A*). Moreover, morphine-treated $CRF_1$ HET and $CRF_1$ KO mice traveled similar distance but showed different social behavior (*Figure 3D*, *Figure 3—figure supplement 1*). Finally, overall mice traveled more distance during the habituation than during the sociability phase, an effect usually observed in the three-chamber test (*Piccin and Contarino, 2020a*; *Piccin and Contarino, 2020b*).

The PVN is a main source of brain CRF (*Sawchenko et al., 1993*). Moreover, CRF released within the PVN may act on intra-PVN $CRF_1$ receptor-expressing neurons (*Jiang et al., 2019*; *Jiang et al., 2018*). Thus, to further explore the mechanisms of $CRF_1$ receptor-mediated sociability deficits, we examined neuronal responses to morphine in the PVN. We found that morphine consistently elevated the firing frequency of PVN neurons in male and female C57BL/6J mice, and in male $CRF_1$ WT and $CRF_1$ HET mice. Acute morphine treatment increases CRF level in the hypothalamus and HPA axis activity (*Buckingham, 1982*; *Ignar and Kuhn, 1990*). Accordingly, stimulation of presynaptic mu-opioid receptors located on PVN GABA terminals might reduce GABA release and thus elevate PVN CRF activity (*Wamsteeker Cusulin et al., 2013*). Thus, although herein we did not examine CRF expression, it is likely that morphine increased the activity of PVN CRF neurons. We also show that $CRF_1$ receptor antagonism by antalarmin completely eliminated morphine-induced PVN neuronal firing in male, but not in female, mice. Likewise, in male mice genetic inactivation of the $CRF_1$ receptor decreased morphine-induced PVN neuronal firing in a $CRF_1$ gene expression-dependent manner. Sex-linked differences in brain distribution and activity of the CRF system might underlie the present findings. For instance, female rats displayed higher CRF expression in the PVN and in the central nucleus of the amygdala (CeA), as compared to male rats (*Iwasaki-Sekino et al., 2009*). However, using a $CRF_1$ reporter mouse line maintained on a C57BL/6 background, studies showed higher levels of the $CRF_1$ receptor in the PVN of adult (2 months) and old (20–24 months) male mice, as compared to adult and old female mice (*Rosinger et al., 2019*). Interestingly, adult gonadectomy (6 weeks) decreased PVN $CRF_1$ receptor-immunoreactive cells in male, but not in female, mice, indicating a sex-linked modulation of $CRF_1$ receptor expression by gonadal hormones (*Rosinger et al., 2019*). Moreover, female rats showed higher $CRF_1$ receptor-GTP-binding protein ($G_s$) coupling and higher $CRF_1$ receptor cellular internalization than male rats in cortical and locus coeruleus (LC) tissues, respectively (*Bangasser et al., 2010*). Notably, swim stress and transgenic CRF overexpression increased or decreased LC $CRF_1$ receptor cellular internalization in male or female rats and mice, respectively (*Bangasser et al., 2013*; *Bangasser et al., 2010*). Thus, it is possible that sex-linked differences in PVN $CRF_1$ receptor expression, $CRF_1$ receptor intracellular signaling or cellular compartmentalization contributed to the sex-linked behavioral and brain effects of antalarmin reported herein. However, more studies are warranted to address the latter hypotheses.

The present immunohistochemistry studies showed that, in male and female C57BL/6J mice, approximately half of the patched PVN cells expressed both OXY and AVP. However, in male mice a relatively large portion of the stained cells expressed AVP, but not OXY. In net contrast, in female mice a large portion of the stained cells expressed OXY, but not AVP. The latter sex differences resonate with previous studies. Indeed, AVP- or OXY-positive neurons were shown to be more numerous in the PVN of male or female animals, respectively, in a variety of species, including humans (*Dumais and Veenema, 2016*). Interestingly, herein morphine disrupted sociability but increased the firing frequency of PVN neurons expressing OXY and/or AVP. At first glance, the present results might seem at odds with the alleged prosocial role for OXY systems. However, PVN OXY neurons extensively project to several brain areas where they may differentially modulate social behavior in a brain site-specific manner (*Jurek and Neumann, 2018*). For instance, genetically driven activation or inhibition of PVN OXY neurons projecting to the ventral tegmental area (VTA), respectively, increased or decreased social interaction in male mice (*Hung et al., 2017*). In contrast, OXY infusion into the bed nucleus of the stria terminalis (BNST) dose-dependently decreased social approach in both male and female California mice (*Duque-Wilckens et al., 2020*). Moreover, both pharmacological antagonism of OXY receptors and genetic inhibition of OXY synthesis within the BNST attenuated social defeat

stress-induced reduction of social interaction in female California mice, further indicating a negative modulation of social behavior by BNST OXY activity (*Duque-Wilckens et al., 2020*; *Duque-Wilckens et al., 2018*). CRF$_1$ receptor mRNA and OXY mRNA were shown to co-localize in PVN neurons in male rats (*Arima and Aguilera, 2000*). Also, PVN CRF$_1$ receptor-expressing neurons are thought to make bidirectional connections with PVN OXY-expressing neurons, suggesting intra-PVN CRF–OXY interactions relevant to stress responses and social behavior (*Jiang et al., 2019*). Thus, activation of PVN CRF$_1$ receptors by morphine-induced CRF release might modulate the activity of PVN OXY neurons projecting to the VTA or the BNST and related social behavior. On the other hand, since CRF$_1$ receptors are highly expressed in the VTA, dopamine activity might also be directly modulated by intra-VTA CRF/CRF$_1$ receptor pathways (*Chen et al., 2014*). Accordingly, intra-VTA administration of CRF receptor antagonists attenuated social defeat stress-induced dopamine release in the nucleus accumbens shell (*Boyson et al., 2014*). However, more studies are needed to determine the role for brain site-specific CRF/CRF$_1$ receptor pathways in OXY and dopamine activity underlying the social behavior effects of substances of abuse.

Stressful events strongly activate brain OXY systems (*Jurek and Neumann, 2018*). For instance, male rats exposed to the forced swim or the tail suspension stressor showed increased OXY peptide levels in several brain areas, including the PVN and the SON (*Yan et al., 2014*). Notably, intracerebroventricular injection of an OXY receptor antagonist dose-dependently increased stress-induced immobility, suggesting that OXY activity served to cope with stress (*Yan et al., 2014*). Acute morphine administration may elicit a stress-like state, as revealed by elevated CRF mRNA in the CeA and HPA axis activity in male rats (*Ignar and Kuhn, 1990*; *Maj et al., 2003*). Within this framework, the present results of morphine-induced firing of PVN OXY-positive neurons suggest the presence of a stress-like state, which may disrupt social behavior. Thus, morphine may activate brain CRF systems which, via CRF$_1$ receptors, may increase the activity of PVN OXY neurons in order to counteract stress effects. In contrast, antalarmin completely eliminated morphine-induced firing of PVN OXY-expressing neurons and sociability deficits in male mice. This suggests that pharmacological disruption of the stress-responsive CRF$_1$ receptor confers stress resilience, which per se does not require PVN OXY activity to cope with a stress-like state, leaving unaltered the expression of social behavior. On the other hand, antalarmin did not affect the activity of neurons expressing AVP, but not OXY, in male mice. Accordingly, prior work indicated a minor role for AVP in stress resilience and sociability, as compared to OXY (*Lukas et al., 2011*; *Neumann and Landgraf, 2012*). However, PVN-targeted genetic or pharmacological studies are needed to determine the role for PVN CRF$_1$ receptors in opiate-induced sociability deficits and related PVN OXY activity. Heroin self-administration may up-regulate CRF$_1$ receptor mRNA level in VTA dopamine neurons in male rats (*Galaj et al., 2023*). Nevertheless, to our knowledge, to date no studies have investigated the effect of acute morphine administration upon PVN CRF$_1$ receptor expression. Thus, together with PVN-targeted genetic or pharmacological manipulation of CRF$_1$ receptor activity, assessing CRF$_1$ receptor expression in the PVN might help to understand the brain substrates of opiate-induced disruption of social behavior. Finally, in female mice antalarmin affected neither morphine-induced sociability deficits nor firing of OXY/AVP- or OXY-expressing neurons, revealing a sex-specific role for the CRF$_1$ receptor in opiate-induced activity of brain OXY systems and related social behavior.

In summary, herein we provide initial evidence of a major, sex-linked, role for the CRF$_1$ receptor in social behavior and brain alterations induced by morphine. Indeed, disruption of CRF$_1$ receptor function consistently eliminated morphine-induced sociability deficits and PVN neuronal firing in male, but not in female, mice. These findings suggest that inhibition of CRF$_1$ receptor activity may relieve severe social behavior deficits commonly observed in OUD patients. Moreover, the present results point out to sex as a critical biological variable of studies assessing novel treatments for substance use disorders.

## Materials and methods
### Animals
Male and female C57BL/6J mice were bred in-house and derived from mice originally purchased from Janvier Labs (Le Genest-Saint-Isle, France). Male and female CRF$_1$ WT, CRF$_1$ HET and CRF$_1$ KO mice previously generated on a mixed C57BL/6Jx129 background were bred in-house from mating CRF$_1$

HET mice and genotype identified by PCR analysis of tail DNA (*Smith et al., 1998*). The colony room (22 ± 2°C, relative humidity: 50–60%) was maintained on a 12-hr light/dark cycle (lights on at 08:00). Mice were housed in groups of 2–4 in transparent polycarbonate cages (29.5 × 11.5 × 13 cm, *L × W × H*) containing bedding and a cotton nestlet (SAFE, Augy, France) and were 12–28 weeks old at testing. Standard laboratory food (3.3 kcal/g; SAFE, Augy, France) and water were available ad libitum. All studies were conducted in accordance with the European Communities Council Directive 2010/63/EU, were approved by the local Animal Care and Use Committee and complied with the ARRIVE Guidelines (*Kilkenny et al., 2010*).

## Three-chamber sociability task

The three-chamber task allowed the study of the preference for an unfamiliar same-sex conspecific versus an object and was carried out as previously reported (*Piccin et al., 2022b*). The three-chamber apparatus was a rectangular box (60 × 40 × 20 cm, *L × W × H*) divided in three equal chambers and made of dark grey polypropylene. Dividing transparent Plexiglas walls had small squared doors (8 × 8 cm) that could be manually opened and closed. The central chamber was empty and each side chamber contained a round wire cage (12 cm diameter, 14 cm high, with bars spaced 1 cm apart) placed in one half-portion of the chamber. The three-chamber test was conducted during the light phase of the 12-hr light/dark cycle and light intensity in the apparatus was ~10 lux. The subject mice were handled 1 min/day during the three days preceding the three-chamber experiment. On the fourth day, C57BL/6J mice were treated with either vehicle or antalarmin (20 mg/kg) and returned to their home-cage. One hour later, they were treated with either saline or morphine (2.5 mg/kg) and immediately tested in the three-chamber task (*Figure 1A*). We did not monitor the estrous cycle in female mice. The normal estrous cycle of laboratory mice is 4–5 days in length, and it is divided into four phases (proestrus, estrus, metestrus, and diestrus). The three-chamber experiments were generally carried out over a 5-day period, thus spanning across the entire estrous cycle. In particular, on each test day approximately the same number of mice was assigned to each experimental group. Thus, within each group the number of female mice tested on each phase of the estrous cycle was likely similar. Moreover, studies indicated no significant difference over different phases of the estrous cycle in social interaction, anxiety- and anhedonia-like behavioral tests in C57BL/6J female mice (*Zeng et al., 2023*; *Zhao et al., 2021*). CRF$_1$ WT, CRF$_1$ HET and CRF$_1$ KO mice were treated with either saline or morphine (0.625 mg/kg) and immediately tested in the three-chamber task (*Figure 3A*). The three-chamber test consisted of three phases: pre-habituation, habituation, and sociability. During the pre-habituation phase, the subject mouse was confined to the central chamber for 5 min. Then, the doors were opened and the mouse could freely explore the three chambers and the empty wire cages for 10 min (habituation phase). During the subsequent 10 min, the subject mouse could freely explore the entire apparatus with one wire cage containing an unfamiliar same-sex mouse and the other an object, that is, a plastic bottle cap (sociability phase). The unfamiliar mice were C57BL/6J mice sex- and age-matched with the subject mice. During the 3 days preceding testing, the unfamiliar mice were handled 1 min/day and habituated to the wire cages for 10 min/day, with the wire cage habituation taking place in the three-chamber apparatus on the second and third day. The position (left- or right-side chamber) of the unfamiliar mouse was counter-balanced within each experimental group. Between each tested mouse, the apparatus was cleaned with water and the wire cages with 70% ethanol and then water. Videos were acquired and analyzed with a home-made tracking system. In particular, time (s) spent by the tested mouse in the ROIs (side half-chambers) containing the wire cages was taken as a measure of sociability (*Figure 1A*). Indeed, our prior studies reliably demonstrated that the latter measure positively correlated with the number of nose-to-nose contacts with the unfamiliar mouse (*Piccin and Contarino, 2020b*). Moreover, ratio of time spent in the ROI containing the unfamiliar mouse positively correlated with the ratio of time spent with the nose in the wire cage containing the unfamiliar mouse (*Piccin and Contarino, 2020b*). Furthermore, to control for locomotor activity, distance (m) traveled throughout the whole apparatus during the habituation and the sociability phases of the test was examined. Sociability ratio was calculated as percentage of time spent in the ROI containing the unfamiliar mouse over the total time spent in both ROIs containing the wire cages.

## Brain slice preparation

C57BL/6J mice were injected with either vehicle or antalarmin (20 mg/kg) and, 1 hr later, with either saline or morphine (2.5 mg/kg). CRF$_1$ receptor-deficient mice were just injected with either saline or morphine (0.625 mg/kg). Ten minutes after saline or morphine administration, mice were anesthetized by intraperitoneal (i.p.) injection of ketamine (100 mg/kg)/xylazine (10 mg/kg) until reflexes to tail- or toe-pinching were lost. Before brain removal, animals were intracardially perfused with an ice-cold bubbled (95% O$_2$/5% CO$_2$) sucrose-based saline solution containing (in mM): NaH$_2$PO$_4$ 1.25, KCl 2.5, CaCl$_2$ 0.5, MgSO$_4$ 10, D-glucose 10, NaHCPO$_3$ 26. Brains were rapidly removed and 300 μm coronal slices containing the PVN were cut using a vibroslicer (Leica VT100S, Leica Biosystems, Germany). Slices were then allowed to recover for at least 1 hr at 30°C in a holding chamber filled with oxygen-ated (95% O$_2$/5% CO$_2$) artificial cerebrospinal fluid (aCSF) composed of (in mM): NaCl 126, KCl 2.5, CaCl$_2$ 2, MgSO$_4$ 2, NaH$_2$PO$_4$ 1.25, NaHCO$_3$ 26, glucose 10 (pH 7.3, 290 mOsm).

## Electrophysiology studies

Cell-attached patch-clamp recordings from PVN neurons were made at room temperature in voltage-clamp conditions under continuous perfusion of oxygenated aCSF composed of (in mM): NaCl 126, KCl 3, CaCl$_2$ 1.6, MgSO$_4$ 1.5, NaH$_2$PO$_4$ 1.25, NaHCO$_3$ 26, glucose 10. Throughout recordings, GABAergic and glutamatergic inputs were blocked with gabazine (1 μM) and 10 μM of the NMDA and non-NMDA receptor antagonists D(−)-2-amino5-phosphonopentanoic acid (AP5) and 6,7-dinitroquinoxaline-2,3 (1H,4H) dione (DNQX). Neurons were visualized with an upright Nikon Eclipse FN1 microscope (Nikon, Japan) with infrared illumination. Recording borosilicate electrodes were filled with an internal solution containing K-Gluconate 120 mM, KCl 20 mM, MgCl$_2$ 1.3 mM, EGTA 1 mM, HEPES 10 mM, CaCl$_2$ 0.1 mM, GTP 0.03 mM, cAMP 0.1 mM, leupeptine 0.01 mM, D-Mannitol 77 mM, and Na 2 ATP 3 mM (pH 7.3). Moreover, biocytin 0.1% was added to the internal solution in order to post-visualize recorded neurons. Data were collected online with a Multiclamp 700B amplifier (Molecular Devices, USA) and acquired with Axograph X software (Axograph, Australia). Electrophysiology recordings were analyzed offline using the Axograph X software.

## Immunohistochemistry and imaging

The phenotype of the patched and recorded cells was assessed by immunohistochemistry. After elec-trophysiological recording, slices were fixed with 4% paraformaldehyde overnight at 4°C. Biocytin was then revealed with FITC-Streptavidin (1/300, Vector Laboratories). OXY and AVP immunohistochemical labeling was performed at the same time using as first antibodies mouse anti-OXY monoclonal anti-body (1/1000, Millipore MAB5296) and T-5048 Guinea pig anti (Arg[8])-vasopressin antibody (1/1000, BMA biomedicals). Alexa fluor 488 goat anti-mouse IgG (1/500, Life technology) and Alexa fluor 647 donkey anti-guinea pig IgG (1/500, Life technology) were used as secondary antibodies. Immunostain-ings were acquired using a confocal Zeiss LSM900 microscope. Serial optical sections were obtained at a Z-step of 1.2 μm and imaged using an objective 10× or 20×/1.00 numerical aperture.

## Drugs

Antalarmin hydrochloride (20 mg/kg; TOCRIS, Lille, France) was dissolved in acidified saline (pH ~2.5) and injected per os (p.o.) by gavage. Morphine hydrochloride (0.625 or 2.5 mg/kg; Francopia, Gentilly, France) was dissolved in physiological saline and injected i.p. Control mice were injected p.o. or i.p. with the appropriate vehicle (acidified or physiological saline) and volume of administration was always 10 ml/kg. The morphine dose was chosen based on our prior studies showing that morphine (2.5 mg/kg, i.p.) impaired sociability in male and female C57BL/6J mice, without affecting locomotor activity (*Piccin et al., 2022b*). Also, the antalarmin dose and route of administration were chosen based on our prior reports of behavioral effects of antalarmin (20 mg/kg) administered p.o. (*Contarino et al., 2017*; *Ingallinesi et al., 2012*; *Piccin and Contarino, 2020b*). Notably, the oral route of administration for antalarmin was chosen for its translational relevance, as it could be easily employed in clinical trials assessing the therapeutic value of pharmacological CRF$_1$ receptor antagonists.

## Statistical analysis

Each mouse was assigned a unique identification number that was used to conduct blind testing and data analysis. To prevent strong initial preferences from biasing the three-chamber sociability results,

animals exploring each ROI containing the wire cage for more than 80% (or less than 20%) of the total time spent in both ROIs during the habituation phase (10 min) were excluded from data analysis. The number of animals excluded within each experimental group is reported in *Supplementary file 1a, b*. For simplification and illustration purposes, data obtained in male and female mice were reported on separate figures. Thus, within each sex, the three-way repeated measures ANOVA with pretreatment (vehicle vs. antalarmin) and treatment (saline vs. morphine) as between-subjects factors and side (mouse vs. object) or test phase (habituation *vs.* sociability) as a within-subject factor was used to analyze time spent in the ROIs or distance traveled during the three-chamber test by C57BL/6J mice. A three-way repeated measures ANOVA with genotype ($CRF_1$ WT vs. $CRF_1$ HET vs. $CRF_1$ KO) and treatment (saline vs. morphine) as between-subjects factors and side (mouse vs. object) or test phase (habituation vs. sociability) as a within-subject factor was used to analyze time spent in the ROIs or distance traveled during the three-chamber test by $CRF_1$ receptor-deficient mice. The two-way ANOVA with pretreatment (vehicle vs. antalarmin) or genotype ($CRF_1$ WT vs. $CRF_1$ HET vs. $CRF_1$ KO) and treatment (saline vs. morphine) as between-subjects factors was used to analyze sociability ratio and the firing frequency (Hz) results of the electrophysiology studies. Sociability ratio and firing frequency displayed by C57BL/6J mice were also examined by a three-way ANOVA with sex (males vs. females), pretreatment (vehicle vs. antalarmin) and treatment (saline vs. morphine) as between-subjects factors. The accepted value for significance was $p < 0.05$. Following significant interaction effects, the Newman–Keuls post hoc test was used for individual group comparisons. Statistical analyses were performed using the Statistica software (Version 10). Data graphs were created using GraphPad Prism and Adobe Illustrator.

## Acknowledgements

The authors would like to thank Dr Philippe Ciofi (INSERM U1215) for the precious help with the oxytocin and vasopressin studies. This study was supported by the *Fondation pour la Recherche Médicale* (Grant No. DPA20140629794 to AC and JB), the *Agence Nationale de la Recherche* (Grant No. ANR-21-CE37-0019-01 to AC), the University of Bordeaux, and the *Centre National de la Recherche Scientifique* (CNRS), France. Funding sources had no further role in study design, in the collection, analysis, and interpretation of data, in the writing of the report, and in the decision to submit the paper for publication.

## Additional information

### Funding

| Funder | Grant reference number | Author |
| --- | --- | --- |
| Fondation pour la Recherche Médicale | DPA20140629794 | Jérôme M Baufreton Angelo Contarino |
| Agence Nationale de la Recherche | ANR-21-CE37-0019-01 | Angelo Contarino |

The funders had no role in study design, data collection and interpretation, or the decision to submit the work for publication.

### Author contributions

Alessandro Piccin, Data curation, Formal analysis, Investigation, Methodology, Writing – original draft; Anne-Emilie Allain, Data curation, Formal analysis, Investigation, Methodology; Jérôme M Baufreton, Data curation, Formal analysis, Investigation; Sandrine S Bertrand, Data curation, Formal analysis, Methodology; Angelo Contarino, Conceptualization, Resources, Data curation, Formal analysis, Supervision, Funding acquisition, Validation, Investigation, Methodology, Writing – original draft, Project administration, Writing – review and editing

### Author ORCIDs

Alessandro Piccin ⓘ https://orcid.org/0000-0001-9566-3808
Jérôme M Baufreton ⓘ https://orcid.org/0000-0002-2623-6375

Sandrine S Bertrand https://orcid.org/0000-0002-3020-7980
Angelo Contarino https://orcid.org/0000-0002-7286-6941

## Ethics

All studies were conducted in accordance with the European Communities Council Directive 2010/63/EU, were approved by the local Animal Care and Use Committee and complied with the ARRIVE Guidelines (Kilkenny et al., 2010. This reference is reported in the manuscript).

Reviewer #1 (Public review): https://doi.org/10.7554/eLife.100849.3.sa1
Author response https://doi.org/10.7554/eLife.100849.3.sa2

## Additional files

### Supplementary files
Supplementary file 1. Number of animals used, number of cells patched and recorded and statistical analyses. (**a–c**) Number of animals used and cells patched and recorded. (**d**) Statistical analysis of the three-chamber sociability test in C57BL/6J mice. (**e**) Statistical analysis of locomotor activity displayed by C57BL/6J mice during the three-chamber sociability test. (**f**) Statistical analysis of the three-chamber sociability test in $CRF_1$ receptor-deficient mice. (**g**) Female $CRF_1$ WT and $CRF_1$ HET mice fail to perform in the three-chamber task for sociability. (**h**) Statistical analysis of neuronal firing in C57BL/6J mice.

MDAR checklist

### Data availability
All of the data are available as a Dryad dataset titled 'Piccin et al. eLife 2024 for Dryad' and can be accessed using the following digital object identifier: https://doi.org/10.5061/dryad.5hqbzkhgj.

The following dataset was generated:

| Author(s) | Year | Dataset title | Dataset URL | Database and Identifier |
|---|---|---|---|---|
| Contarino A | 2025 | Piccin et al. eLife 2024 for Dryad | https://doi.org/10.5061/dryad.5hqbzkhgj | Dryad Digital Repository, 10.5061/dryad.5hqbzkhgj |

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
