## [Editor Report · eLife Assessment]

The revised report provides **valuable** findings for the field, suggesting a relationship between CRF1 receptors, sociability deficits in morphine-treated male mice yet not females, and a potential mechanism involving oxytocin neurons in the paraventricular nucleus of the hypothalamus. Generally, the strength of evidence is **solid** in terms of the methods, data, and analyses. This work will be of interest to those interested in social behavior and addiction.

---

## [Referee Report · Reviewer #1 (Public review)]

Summary:

The use of antalarmin, a selective CRF1 receptor antagonist, prevents the deficits in sociability in (acutely) morphine-treated males, but not in females. In addition, cell attached experiments show a rescue to control levels of the morphine-induced increased firing in PVN neurons from morphine-treated males. Similar results are obtained in CRF receptor 1-/- male mice, confirming the involvement of CRF receptor 1-mediated signaling in both sociability deficits and neuronal firing changes in morphine-treated male mice.

Strengths:

In the revised version of the paper the authors respond to some reviewers's points with a new statistical analysis of behavioral data and a new discussion of previous literature.

Weaknesses:

Following reviewers' comments, the authors provided mechanistic insights of their findings with new experiments.

---

## [Author Response]

The following is the authors’ response to the original reviews.

**Public Reviews:**

**Reviewer #1 (Public review):**
Summary:The use of antalarmin, a selective CRF1 receptor antagonist, prevents the deficits in sociability in (acutely) morphine-treated males, but not in females. In addition, cell-attached experiments show a rescue to control levels of the morphine-induced increased firing in PVN neurons from morphine-treated males. Similar results are obtained in CRF receptor 1-/- male mice, confirming the involvement of CRF receptor 1-mediated signaling in both sociability deficits and neuronal firing changes in morphine-treated male mice.Strengths:The experiments and analyses appear to be performed to a high standard, and the manuscript is well written and the data clearly presented. The main finding, that CRF-receptor plays a role in sociability deficits occurring after acute morphine administration, is an important contribution to the field.Weaknesses:The link between the effect of pharmacological and genetic modulation of CRF 1 receptor on sociability and on PVN neuronal firing, is less well supported by the data presented. No evidence of causality is provided.Major points:(1) The results of behavioral tests and the neural substrate are purely correlative. To find causality would be important to selectively delete or re-express CRF1 receptor sequence in the VPN. Re-expressing the CRF1 receptor in the VPN of male mice and testing them for social behavior and for neuronal firing would be the easier step in this direction.

We agree with this comment and have acknowledged that further studies, such as genetic or pharmacological inactivation of CRF_1_ receptors selectively in the paraventricular nucleus of the hypothalamus (PVN), are warranted to address this issue (page 17, line 25 to page 18, line 1).

We would also like to mention that our manuscript title intentionally presented our findings separately without implying causality. Our idea was simply to pair the behavioral data to neural activity within a network of interest, i.e., the PVN CRF-oxytocin (OXY)/arginine-vasopressin (AVP) network, which is thought to play a critical role at the interface of substance use disorders and social behavior. Accordingly, we previously reported that genetic CRF_2_ receptor deficiency reliably eliminated sociability deficits and hypothalamic OXY and AVP expression induced by cocaine withdrawal (Morisot et al., 2018). Thus, the present manuscript reliably shows that CRF_1_ receptor-mediated effects of acute morphine administration upon social behavior are consistently mirrored by neural activity changes within the PVN, and particularly within its OXY^+^/AVP^+^ neuronal populations. In addition, we demonstrate that the latter effects are sex-linked, which is in line with previous reports of sex-biased CRF_1_ receptor roles in rodents (Rosinger et al., 2019; Valentino et al., 2013) and humans (Roy et al., 2018; Weber et al., 2016).

(2) It would be interesting to discuss the relationship between morphine dose and CRF1 receptor expression.

We are not aware of studies reporting CRF_1_ receptor expression following acute morphine administration. However, repeated heroin self-administration was shown to increase CRF_1_ receptor expression in the ventral tegmental area (VTA). We have mentioned the latter study in the present revised version of our manuscript at page 18, lines 1-2.

(3) It would be important to show the expression levels of CRF1 receptors in PVN neurons in controls and morphine-treated mice, both males and females.

We agree with this reviewer comment and, in the present version of the manuscript, have mentioned that examination of CRF_1_ receptor expression in the PVN might help to understand the brain mechanisms underlying morphine effects upon social behavior (page 18, lines 2-6). Moreover, at page 15, lines 11-19 we have mentioned studies showing higher levels of the CRF_1_ receptor in the PVN of adult (2 months) and old (20-24 months) male mice, as compared to adult and old female mice (Rosinger et al., 2019). Thus, differences in PVN CRF_1_ receptor expression between male and female mice might underlie the sex-linked effects of CRF_1_ receptor antagonism by antalarmin reported in our manuscript.

(4) It would be important to discuss the mechanisms by which CRF1 receptor controls the firing frequency of APV+/OXY+ neurons in the VPN of male mice.

Using the in situ hybridization technique, studies reported relatively low expression of the CRF_1_ receptor in the PVN (Van Pett et al., 2000). However, more recent studies using genetic approaches identified a substantial population of CRF_1_ receptor-expressing neurons within the PVN (Jiang et al., 2019, 2018). These CRF_1_ receptor-expressing neurons are believed to respond to local CRF release and likely form bidirectional connections with both CRF and OXY+/AVP+ neurons (Jiang et al., 2019, 2018). Thus, one proposed mechanism of action is that morphine increases intra-PVN release of CRF, which may act on intra-PVN CRF_1_ receptor-expressing neurons. The latter neurons might in turn influence the activity of PVN OXY+/AVP+ neurons, which largely project to the VTA and the bed nucleus of the stria terminalis (BNST) to modulate social behavior. Within this framework, pharmacological or genetic inactivation of CRF_1_ receptors might deregulate the activity of intra-PVN CRF-OXY/AVP interactions and thus interfere with opiate-induced social behavior deficits. In particular, the latter phenomenon might be more pronounced in male mice since they express more CRF_1_ receptor-positive neurons in the PVN, as compared to female mice (Rosinger et al., 2019). The putative mechanisms of action described herein are also mentioned at page 16, line 12 to page 17, line 7 of the present revised version of the manuscript.

Minor points:(1) The phase of the estrous cycles in which females are analyzed for both behavior and electrophysiology should be stated.

The normal estrous cycle of laboratory mice is 4-5 days in length, and it is divided into four phases (proestrus, estrus, metestrus and diestrus). The three-chamber experiments were generally carried out over a 5-day period, thus spanning across the entire estrous cycle. In particular, on each test day approximately the same number of mice was assigned to each experimental group. Thus, within each group the number of female mice tested on each phase of the estrous cycle was likely similar. Moreover, except for firing frequency displayed by vehicle/morphine-treated mice, female and male mice showed similar results *variability*, indicating a marginal role for the estrous cycle in the spread of data. We would also like to mention relatively recent studies indicating no significant difference over different phases of the estrous cycle in the social interaction test as well as in anxiety-like and anhedonia-like behavioral tests in C57BL/6J female mice (Zhao et al., 2021). Accordingly, similar findings were also reported by other authors who found no difference across the diestrus and estrus phases of the estrous cycle in C57BL/6J female mice tested in behavioral assays of anxiety-like, depression-like and social interaction (Zeng et al., 2023).

A paragraph has been added to page 20, lines 1-9 of the present version of the manuscript to explain why we did not monitor the estrous cycle in female mice.

(2) It would be important to show the statistical analysis between sexes.

Following this reviewer comment, we examined the sociability ratio results by a three-way ANOVA with sex (males *vs.* females), pretreatment (vehicle *vs.* antalarmin) and treatment (saline *vs.* morphine) as between-subjects factors. The latter analysis revealed an almost significant sex X pretreatment X treatment interaction effect (F_1,53_=3.287, P=0.075), which could not allow for post-hoc individual group comparisons. Nevertheless, Newman-Keuls post-hoc comparisons revealed that male mice treated with antalarmin/morphine showed higher sociability ratio than female mice treated with antalarmin/morphine (P<0.05). The latter statistical results have been added to the present revised version of the manuscript at page 7, lines 2-8.

We also examined neuronal firing frequency by a three-way ANOVA with sex (males *vs.* females), pretreatment (vehicle *vs.* antalarmin) and treatment (saline *vs.* morphine) as between-subjects factors. Analysis of firing frequency of all of the recorded cells in C57BL/6J mice revealed a sex X pretreatment X treatment interaction effect (F_1,195_=4.765, P<0.05). Newman-Keuls post-hoc individual group comparisons revealed that male mice treated with vehicle/morphine showed higher firing frequency than all other male and female groups (P<0.0005). Moreover, male mice treated with antalarmin/morphine showed lower firing frequency than male mice treated with vehicle/morphine (P<0.0005). In net contrast, female mice treated with antalarmin/morphine did not differ from female mice treated with vehicle/morphine (P=0.914). The latter statistical results have been added to the present revised version of the manuscript at page 8, lines 4-12. Finally, similar results were obtained following the three-way ANOVA (sex X pretreatment X treatment) of firing frequency recorded in the subset of neurons co-expressing OXY and AVP (data not shown).

Thus, sex-linked responses to morphine were detected also by three-way ANOVAs including sex as a variable. However, in the revised version of the manuscript we did not include novel figures combining the two sexes because it would have been largely redundant with the figures already reported, especially with Figure 1D, Figure 1G, Figure 2B and Figure 2D.

**Reviewer #2 (Public review):**
This manuscript reports a series of studies that sought to identify a biological basis for morphine-induced social deficits. This goal has important translational implications and is, at present, incompletely understood in the field. The extant literature points to changes in periventricular CRF and oxytocin neurons as critical substrates for morphine to alter social behavior. The experiments utilize mice, administered morphine prior to a sociability assay. Both male and female mice show reduced sociability in this procedure. Pretreatment with the CRF1 receptor antagonist, antalarmin, clearly abolished the morphine effect in males, and the data are compelling. Consistently, CRF1-/- male mice appeared to be spared of the effect of morphine (while wild-type and het mice had reduced sociability). The same experiment was reported as non-feasible in females due to the effect of dose on exploratory behavior per se. Seeking a neural correlate of the behavioral pharmacology, acute cell-attached recordings of PVN neurons were made in acute slices from mice pretreated with morphine or anatalarmin. Morphine increased firing frequencies, and both antalarmin and CRF1-/- mice were spared of this effect. Increasing confidence that this is a CRF1 mediated effect, there is a gene deletion dose effect where het's had an intermediate response to morphine. In general, these experiments are well-designed and sufficiently powered to support the authors' inferences. A final experiment repeated the cell-attached recordings with later immunohistochemical verification of the recorded cells as oxytocin or vasopressin positive. Here the data are more nuanced. The majority of sampled cells were positive for both oxytocin and vasopressin, in cells obtained from males, morphine pretreatment increased firing in this population and was CRF1 dependent, however in females the effect of morphine was more modest without sensitivity to CRF1. Given that only ~8 cells were only immunoreactive for oxytocin, it may be premature to attribute the changes in behavior and physiology strictly to oxytocinergic neurons.In sum, the data provide convincing behavioral pharmacological evidence and a regional (and possibly cellular) correlation of these effects suggesting that morphine leads to sociality deficits via CRF interacting with oxytocin in the hypothalamus. While this hypothesis remains plausible, the current data do not go so far as directly testing this mechanism in a site or cell-specific way.

We agree with this reviewer’s comment and acknowledge that further studies are needed to better understand the neural substrates of CRF_1_ receptor-mediated sociability deficits induced by morphine. This has been mentioned at page 17, line 25 to page 18, line 6 of the present revised version of the manuscript.

With regard to the presentation of these data and their interpretation, the manuscript does not sufficiently draw a clear link between mu-opioid receptors, their action on CRF neurons of the PVN, and the synaptic connectivity to oxytocin neurons. Importantly, sex, cell, and site-specific variations in the CRF are well established (see Valentino & Bangasser) yet these are not reviewed nor are hypotheses regarding sex differences articulated at the outset. The manuscript would have more impact on the field if the implications of the sex-specific effects evident here were incorporated into a larger literature.

At page 15, line 19 to page 16, line 2 of the present version of the manuscript, we have mentioned prior studies reporting differences in CRF_1_ receptor signaling or cellular compartmentalization between male and female rodents (Bangasser et al., 2013, 2010). However, the latter studies were conducted in cortical or locus coeruleus brain tissues. Thus, more studies are needed to examine CRF_1_ receptor signaling or cellular compartmentalization in the PVN and their relationship to the sex-linked results reported in our manuscript.

With regards to the model proposed in the discussion, it seems that there is an assumption that ip morphine or antalarmin have specific effects on the PVN and that these mediate behavior - but this is impossible to assume and there are many meaningful alternatives (for example, both MOR and CRF modulation of the raphe or accumbens are worth exploration).

We focused our discussion on PVN OXY/AVP systems because our electrophysiology studies examined neurons expressing OXY and/or AVP in this brain area. However, we understand that other brain areas/systems might mediate the effect of systemic administration of the CRF_1_ receptor antagonist antalarmin or whole-body genetic disruption of the CRF_1_ receptor upon morphine-induced social behavior deficits. For this reason, at page 16, line 12 to page 17, line 7 of the present version of the manuscript we have mentioned the possible involvement of BNST OXY or VTA dopamine systems in the CRF_1_ receptor-mediated social behavior effects of morphine reported herein. Indeed, literature suggests important CRF-OXY and CRF-dopamine interactions in the BNST and the VTA, which might be relevant to the expression of social behavior. Nevertheless, to date the implication of the latter brain systems interactions in social behavior alterations induced by substances of abuse remains to be elucidated.

While it is up to the authors to conduct additional studies, a demonstration that the physiology findings are in fact specific to the PVN would greatly increase confidence that the pharmacology is localized here. Similarly, direct infusion of antalarmin to the PVN, or cell-specific manipulation of OT neurons (OT-cre mice with inhibitory dreadds) combined with morphine pre-exposure would really tie the correlative data together for a strong mechanistic interpretation.

We agree with this reviewer’s comment that the suggested experiments would greatly increase the understanding of the brain mechanisms underlying the social behavior deficits induced by opiate substances. We have acknowledged this at page 17, line 25 to page 18, line 6.

Because the work is framed as informing a clinical problem, the discussion might have increased impact if the authors describe how the acute effects of CRF1 antagonists and morphine might change as a result of repeated use or withdrawal.

Prior studies reported behavioral and neuroendocrine (hypothalamus-pituitary-adrenal axis) effects of chronic systemic administration of CRF_1_ receptor antagonists, such as R121919 and antalarmin (Ayala et al., 2004; Dong et al., 2018). However, to our knowledge, no studies have directly compared the behavioral effects of acute *vs.* repeated administration of CRF_1_ receptor antagonists. We previously reported that *acute* administration of antalarmin increased the expression of somatic opiate withdrawal in mice, indicating that this compound is effective following withdrawal from repeated morphine administration (Papaleo et al., 2007). Nevertheless, further studies are needed to specifically address this reviewer’s comment.

**Reviewer #3 (Public review):**
Summary:In the current manuscript, Piccin et al. identify a role for CRF type 1 receptors in morphine-induced social deficits using a 3-chamber social interaction task in mice. They demonstrate that pre-treatment with a CRFR1 antagonist blocks morphine-induced social deficits in male, but not female, mice, and this is associated with the CRF R1 antagonist blocking morphine-induced increases in PVN neuronal excitability in male but not female mice. They followed up by using a transgenic mouse CRFR1 knockout mouse line. CRFR1 genetic deletion also blocked morphine-induced social deficits, similar to the pharmacological approach, in male mice. This was also associated with morphine-induced increases in PVN neuronal excitability being blocked in CRFR1 knockout mice. Interestingly they found that the pharmacological antagonism of the CRFR1 specifically blocked morphine-induced increases in oxytocin/AVP neurons in the PVN in male mice.Strengths:The authors used both male and female mice where possible and the studies were fairly well controlled. The authors provided sufficient methodological detail and detailed statistical information. They also examined measures of locomotion in all of the behavioral tasks to separate changes in sociability from overall changes in locomotion. The experiments were well thought out and well controlled. The use of both the pharmacological and genetic approaches provides converging lines of evidence for the role of CRFR1 in morphine-induced social deficits. Additionally, they have identified the PVN as a potential site of action for these CRFR1 effects.Weaknesses:While the authors included both sexes they analyzed them independently. This was done for simplicity's sake as they have multiple measures but there are several measures where the number of factors is reduced and the inclusion of sex as a factor would be possible.

Please, see above our response to the same comment made by Reviewer 1.

Additionally, single doses of both the CRFR1 antagonist and morphine are used within an experiment without justification for the doses. In fact, a lower dose of morphine was needed for the genetic CRFR1 mouse line. This would suggest that the dose of morphine being used is likely causing some aversion that may be more present in the females, as they have lower overall time in the ROI areas of both the object and the mouse following morphine exposure.

The morphine dose was chosen based on our prior study showing that morphine (2.5 mg/kg) impaired sociability in male and female C57BL/6J mice, without affecting locomotor activity (Piccin et al., 2022). Also, the antalarmin dose (20 mg/kg) and the route of administration (*per os*) was chosen based on our prior studies demonstrating behavioral effects of this CRF_1_ receptor antagonist administered *per os* (Contarino et al., 2017; Ingallinesi et al., 2012; Piccin and Contarino, 2020). This is now mentioned in the “materials and methods” section of the present revised version of the manuscript at page 23, lines 6-13. We also agree with this reviewer that female mice seemed more sensitive to morphine than male mice. Indeed, during the habituation phase of the three-chamber test female mice treated with morphine (2.5 mg/kg) spent less time in the ROIs containing the empty wire cages, as compared to saline-treated female mice (Figure 1E). However, morphine did not affect locomotor activity in female mice (Figure 1-figure supplement 1B), suggesting independency between social approach and ambulation.

As for the discussion, the authors do not sufficiently address why CRFR1 has an effect in males but not females and what might be driving that difference, or why male and female mice have different distribution of PVN cell types during the recordings.

At page 15, line 11 to page 16, line 2, we have mentioned possible mechanisms that might underlie the sex-linked results reported in our manuscript. Moreover, at page 16, lines 6-9 we have mentioned a seminal review reporting sex-linked expression of PVN OXY and AVP in a variety of animal species that is similar to the present results. Nevertheless, as mentioned in the “discussion” section, further studies are needed to elucidate the neural substrates underlying sex-linked effects of opiate substances upon social behavior.

Additionally, the authors attribute their effect to CRF and CRFR1 within the PVN but do not consider the role of extrahypothalamic CRF and CRFR1. While the PVN does contain the largest density of CRF neurons there are other CRF neurons, notably in the central amygdala and BNST, that have been shown to play important roles in the impact of stress on drug-related behavior. This also holds true for the expression of CRFR1 in other regions of the brain, including the VTA, which is important for drug-related behavior and social behavior. The treatments used in the current manuscript were systemic or brain-wide deletion of CRFR1. Therefore, the authors should consider that the effects could be outside the PVN.

Even if they suggest a role for PVN CRF_1_-OXY circuits, we are aware that the present data do not support a direct link between behavior and PVN CRF_1_ receptors. Thus, at page 16, line 12 to page 17, line 7 of the present version of the manuscript we have mentioned some studies showing a role for PVN OXY, BNST OXY or VTA dopamine systems in social behavior. Interestingly, the latter brain systems are thought to interact with the CRF system. However, more studies are warranted to understand the implication of CRF-OXY or CRF-dopamine interactions in social behavior deficits induced by substances of abuse.

**Recommendations for the authors:**

**Reviewer #2 (Recommendations for the authors):**
I commend the authors on crafting a well-written and clear manuscript with excellent figures. Furthermore, the data analysis and rigor are quite high. I have a few suggestions in the order they appear in the manuscript:The introduction has a number of abrupt transitions. For example, the sentence beginning with "Besides," in paragraph 2 jumps from CRF to oxytocin and vasopressin without a transition or justification. In all, vasopressin may be better removed from the introduction. There is sufficient evidence in the literature to support the CRF-OT circuit that might mediate behavioral pharmacology and this should be clearly described in the introduction.

We have added a sentence at page 3, lines 22-23 to introduce possible interactions of the CRF system with other brain systems implicated in social behavior. Also, in the “introduction” section both OXY and AVP systems are mentioned because our electrophysiology studies examined the effect of morphine upon the activity of OXY- and AVP-positive neurons.

Our interest in the PVN CRF-OXY/AVP network also stems from previous findings from our laboratory showing that genetic inactivation of the CRF_2_ receptor eliminated both sociability deficits and increased hypothalamic OXY and AVP expression associated with long-term cocaine withdrawal in male mice (Morisot et al., 2018). Moreover, evidence suggests the implication of AVP systems in opiate effects. In particular, pharmacological antagonism of AVP-V1b receptors decreased the acquisition of morphine-induced conditioned place preference in male C57BL/6N mice housed with morphine-treated mice (Bates et al., 2018).

Throughout the manuscript, it seems that there is an assumption that ip morphine or antalarmin have specific effects on the PVN and that these mediate behavior - this is impossible to assume and there are many meaningful alternatives (for example, both MOR and CRF modulation of the raphe or accumbens are worth exploration). While it is up to the authors to conduct additional studies, a demonstration that the physiology findings are in fact specific to the PVN would greatly increase confidence that the pharmacology is localized here. Similarly, direct infusion of antalarmin to the PVN, or cell-specific manipulation of OT neurons (OT-cre mice with inhibitory dreadds) combined with morphine pre-exposure would really tie the correlative data together for a strong mechanistic interpretation.

We agree that the suggested experiments would greatly increase the understanding of the brain mechanisms underlying the social behavior deficits induced by opiate substances. This has been acknowledged at page 17, line 25 to page 18, line 6 of the present version of the manuscript.

Also in the introduction, the reference to shank3b mice is not the most direct evidence of oxytocin involvement in sociability. It may be helpful to point reviewers to studies with direct manipulation of these populations (Grinevich group, for example).

At page 4, lines 4-6 of the “introduction” section, we have added a sentence to mention a seminal paper by the Grinevich group demonstrating an important role for OXY-expressing PVN parvocellular neurons in social behavior (Tang et al., 2020). Moreover, at page 4, lines 8-10 we have mentioned a recent study showing that targeted chemogenetic silencing of PVN OXY neurons in male rats impaired short- and long-term social recognition memory (Thirtamara Rajamani et al., 2024).

It would be helpful in the figures to indicate which panels contain male or female data.

The sex of the mice is mentioned above each panel of the main and supplemental figures, except for the studies with CRF_1_ receptor-deficient mice wherein only experiments carried out with male mice were illustrated. In the latter case, the sex (male) of the mice is mentioned in the related legend.

The discussion itself departs from the central data in a few ways - the passages suggesting that morphine produces a stress response and that CRF1 antagonists would block the stress state are highly speculative (although testable). The manuscript would have more impact if the sex-specific effects and alternative hypotheses were enhanced in the discussion.

At page 16, line 12 to page 17, line 7 of the “discussion” section, we have suggested that interaction of the CRF system with other brain systems implicated in social behavior (i.e., OXY, dopamine) might underlie the sex-linked CRF_1_ receptor-mediated effects of morphine reported in our manuscript. Also, at page 15, line 19 to page 16, line 2 we have mentioned studies showing sex-linked CRF_1_ receptor signaling and cellular compartmentalization that might be relevant to the present findings. Finally, to further support the notion of morphine-induced PVN CRF activity, at page 15, lines 4-6 we have mentioned a study suggesting that activation of presynaptic mu-opioid receptors located on PVN GABA terminals might reduce GABA release (and related inhibitory effects) onto PVN CRF neurons (Wamsteeker Cusulin et al., 2013). Nevertheless, we believe that more work is needed to better understand the role for the CRF_1_ receptor in opiate-induced stress responses and activity of OXY and dopamine systems implicated in social behavior.

**Reviewer #3 (Recommendations for the authors):**
(1) You should provide justification for the doses selected for treatments and the route of administration for the CRFR1 antagonist, especially for females.

This has been added at page 23, lines 6-13 of the present version of the manuscript. In particular, the doses and routes of administration for morphine and antalarmin used in the present study were chosen based on previous work from our laboratory. Indeed, the intraperitoneal administration of morphine (2.5 mg/kg) impaired social behavior in male and female mice, without affecting locomotor activity (Piccin et al., 2022). Moreover, the oral route of administration for antalarmin was chosen for its translational relevance, as it could be easily employed in clinical trials assessing the therapeutic value of pharmacological CRF_1_ receptor antagonists.

(2) For the electrophysiology data you should include the number of cells per animal that were obtained. It appears that fewer cells from more females were obtained than in males and so the distribution of individual animals to the overall variance may be different between males and females.

The number of cells examined and animals used in the electrophysiology experiments are reported above each panel of the related Figures 2, 3 and 4 as well as in the supplementary files 1b-c. Overall, the number of cells examined in male and female mice was quite similar. Also, the number of male and female mice used was comparable. Standard errors of the mean (SEM) were quite similar across the different male and female groups (Figures 2B and 2D), except for vehicle/morphine-treated male mice. Indeed, in the latter group a considerable number of cells displayed elevated firing responses to morphine, which accounted for the higher spread of the data. Accordingly, as mentioned above, the three-way ANOVA with sex (males *vs.* females), pretreatment (vehicle *vs.* antalarmin) and treatment (saline *vs.* morphine) as between-subjects factors revealed that male mice treated with vehicle/morphine showed higher firing frequency than all other male and female groups (P<0.0005). Finally, a similar pattern of firing frequency was observed also in neurons co-expressing OXY and AVP, wherein vehicle/morphine-treated male mice displayed higher SEM, as compared to all other male and female groups (Figures 4C and 4F). Thus, except for vehicle/morphine-treated mice, distribution of the firing frequency data did not seem to be linked to the sex of the animal.

(3) You should consider using a nested analysis for the slice electrophysiology data as that is more appropriate.

We thank the reviewer for this suggestion. However, after careful consideration, we have decided to keep the current statistical analyses. In particular, given the relatively low variability of our data, we believe that the use of parametric ANOVA tests is appropriate. Moreover, additional details supporting our choice are provided just above in our response to the comment #2.

(4) While it makes sense to not want to directly compare male and female data that results in needing to run a 4-way ANOVA, there are many measures, such as sociability, firing rate, etc., that if including sex as a factor would result in running a 3-way ANOVA and would allow for direct comparison of male and female mice.

Please, see above our response to the same comment made by Reviewer 1. Notably, the results of our new statistical analyses including sex as a variable further support sex-linked effects of the CRF_1_ receptor antagonist antalarmin upon morphine-induced sociability deficits and PVN neuronal firing. Nevertheless, we would like to keep the figures illustrating our findings as they are since it easily allows detecting the observed sex-linked results. Finally, we hope that this reviewer agrees with our choice, which is consistent with the wording of the title (i.e., “in male mice”).

(5) There are grammatical and phrasing issues throughout the manuscript and the manuscript would benefit from additional thorough editing.

We appreciate this reviewer’s feedback. Thus, upon revising, we have carefully edited the manuscript with regard to possible grammatical and phrasing errors. We hope that our changes have made the manuscript clearer in order to facilitate readability by the audience.

(6) The discussion should be edited to include consideration of an explanation for the presence of the effect in male, but not female, mice more clearly. The discussion should also include some discussion as to why the distribution of cell types used in the electrophysiology recordings was different between males and females and whether the distribution of CRFR1 is different between males and females. Lastly, the authors need to include consideration of extrahypothalamic CRF and CRFR1 as a possible explanation for their effects. While they have PVN neuron recordings, the treatments that they used are brain-wide and therefore the possibility that the critical actions of CRFR1 could be outside the PVN.

At page 15, line 11 to page 16, line 2 of the “discussion” section, we have suggested several mechanisms that might underlie the sex-linked behavioral and brain effects of CRF_1_ receptor antagonism reported in our manuscript. With regard to the distribution of cell types examined in the electrophysiology studies, at page 16, lines 6-9 we have mentioned a seminal review reporting sex-linked expression of PVN OXY and AVP in a variety of animal species that is similar to our results. Moreover, at page 18, lines 2-6 we mentioned that more studies are needed to examine PVN CRF_1_ receptor expression in male and female animals, an issue that is still poorly understood. Finally, at page 16, line 12 to page 17, line 7 of the “discussion” section we also suggest that CRF_1_ receptor-expressing brain areas other than the PVN, such as the BNST or the VTA, might contribute to the sex-linked effects of morphine reported in our manuscript. Thus, in agreement with this reviewer’s suggestion, in the present version of the manuscript we have further emphasized the possible implication of CRF_1_ receptor-expressing extrahypothalamic brain areas in social behavior deficits induced by opiate substances.

References

Ayala AR, Pushkas J, Higley JD, Ronsaville D, Gold PW, Chrousos GP, Pacak K, Calis KA, Gerald M, Lindell S, Rice KC, Cizza G. 2004. Behavioral, adrenal, and sympathetic responses to long-term administration of an oral corticotropin-releasing hormone receptor antagonist in a primate stress paradigm. J Clin Endocrinol Metab 89:5729–5737. doi:10.1210/jc.2003-032170

Bangasser DA, Curtis A, Reyes B a. S, Bethea TT, Parastatidis I, Ischiropoulos H, Van Bockstaele EJ, Valentino RJ. 2010. Sex differences in corticotropin-releasing factor receptor signaling and trafficking: potential role in female vulnerability to stress-related psychopathology. Mol Psychiatry 15:877, 896–904. doi:10.1038/mp.2010.66

Bangasser DA, Reyes B a. S, Piel D, Garachh V, Zhang X-Y, Plona ZM, Van Bockstaele EJ, Beck SG, Valentino RJ. 2013. Increased vulnerability of the brain norepinephrine system of females to corticotropin-releasing factor overexpression. Mol Psychiatry 18:166–173. doi:10.1038/mp.2012.24

Bates MLS, Hofford RS, Emery MA, Wellman PJ, Eitan S. 2018. The role of the vasopressin system and dopamine D1 receptors in the effects of social housing condition on morphine reward. Drug Alcohol Depend 188:113–118. doi:10.1016/j.drugalcdep.2018.03.021

Contarino A, Kitchener P, Vallée M, Papaleo F, Piazza P-V. 2017. CRF1 receptor-deficiency increases cocaine reward. Neuropharmacology 117:41–48. doi:10.1016/j.neuropharm.2017.01.024

Dong H, Keegan JM, Hong E, Gallardo C, Montalvo-Ortiz J, Wang B, Rice KC, Csernansky J. 2018. Corticotrophin releasing factor receptor 1 antagonists prevent chronic stress-induced behavioral changes and synapse loss in aged rats. Psychoneuroendocrinology 90:92–101. doi:10.1016/j.psyneuen.2018.02.013

Ingallinesi M, Rouibi K, Le Moine C, Papaleo F, Contarino A. 2012. CRF2 receptor-deficiency eliminates opiate withdrawal distress without impairing stress coping. Mol Psychiatry 17:1283–1294. doi:10.1038/mp.2011.119

Jiang Z, Rajamanickam S, Justice NJ. 2019. CRF signaling between neurons in the paraventricular nucleus of the hypothalamus (PVN) coordinates stress responses. Neurobiol Stress 11:100192. doi:10.1016/j.ynstr.2019.100192

Jiang Z, Rajamanickam S, Justice NJ. 2018. Local Corticotropin-Releasing Factor Signaling in the Hypothalamic Paraventricular Nucleus. J Neurosci 38:1874–1890. doi:10.1523/JNEUROSCI.1492-17.2017

Morisot N, Monier R, Le Moine C, Millan MJ, Contarino A. 2018. Corticotropin-releasing factor receptor 2-deficiency eliminates social behaviour deficits and vulnerability induced by cocaine. Br J Pharmacol 175:1504–1518. doi:10.1111/bph.14159

Papaleo F, Kitchener P, Contarino A. 2007. Disruption of the CRF/CRF1 receptor stress system exacerbates the somatic signs of opiate withdrawal. Neuron 53:577–589. doi:10.1016/j.neuron.2007.01.022

Piccin A, Contarino A. 2020. Sex-linked roles of the CRF1 and the CRF2 receptor in social behavior. J Neurosci Res 98:1561–1574. doi:10.1002/jnr.24629

Piccin A, Courtand G, Contarino A. 2022. Morphine reduces the interest for natural rewards. Psychopharmacology (Berl) 239:2407–2419. doi:10.1007/s00213-022-06131-7

Rosinger ZJ, Jacobskind JS, De Guzman RM, Justice NJ, Zuloaga DG. 2019. A sexually dimorphic distribution of corticotropin-releasing factor receptor 1 in the paraventricular hypothalamus. Neuroscience 409:195–203. doi:10.1016/j.neuroscience.2019.04.045

Roy A, Laas K, Kurrikoff T, Reif A, Veidebaum T, Lesch K-P, Harro J. 2018. Family environment interacts with CRHR1 rs17689918 to predict mental health and behavioral outcomes. Prog Neuropsychopharmacol Biol Psychiatry 86:45–51. doi:10.1016/j.pnpbp.2018.05.004

Tang Y, Benusiglio D, Lefevre A, Hilfiger L, Althammer F, Bludau A, Hagiwara D, Baudon A, Darbon P, Schimmer J, Kirchner MK, Roy RK, Wang S, Eliava M, Wagner S, Oberhuber M, Conzelmann KK, Schwarz M, Stern JE, Leng G, Neumann ID, Charlet A, Grinevich V. 2020. Social touch promotes interfemale communication via activation of parvocellular oxytocin neurons. Nat Neurosci 23:1125–1137. doi:10.1038/s41593-020-0674-y

Thirtamara Rajamani K, Barbier M, Lefevre A, Niblo K, Cordero N, Netser S, Grinevich V, Wagner S, Harony-Nicolas H. 2024. Oxytocin activity in the paraventricular and supramammillary nuclei of the hypothalamus is essential for social recognition memory in rats. Mol Psychiatry 29:412–424. doi:10.1038/s41380-023-02336-0

Valentino RJ, Van Bockstaele E, Bangasser D. 2013. Sex-specific cell signaling: the corticotropin-releasing factor receptor model. Trends Pharmacol Sci 34:437–444. doi:10.1016/j.tips.2013.06.004

Van Pett K, Viau V, Bittencourt JC, Chan RK, Li HY, Arias C, Prins GS, Perrin M, Vale W, Sawchenko PE. 2000. Distribution of mRNAs encoding CRF receptors in brain and pituitary of rat and mouse. J Comp Neurol 428:191–212. doi:10.1002/1096-9861(20001211)428:2<191::aid-cne1>3.0.co;2-u

Wamsteeker Cusulin JI, Füzesi T, Inoue W, Bains JS. 2013. Glucocorticoid feedback uncovers retrograde opioid signaling at hypothalamic synapses. Nat Neurosci 16:596–604. doi:10.1038/nn.3374

Weber H, Richter J, Straube B, Lueken U, Domschke K, Schartner C, Klauke B, Baumann C, Pané-Farré C, Jacob CP, Scholz C-J, Zwanzger P, Lang T, Fehm L, Jansen A, Konrad C, Fydrich T, Wittmann A, Pfleiderer B, Ströhle A, Gerlach AL, Alpers GW, Arolt V, Pauli P, Wittchen H-U, Kent L, Hamm A, Kircher T, Deckert J, Reif A. 2016. Allelic variation in CRHR1 predisposes to panic disorder: evidence for biased fear processing. Mol Psychiatry 21:813–822. doi:10.1038/mp.2015.125

Zeng P-Y, Tsai Y-H, Lee C-L, Ma Y-K, Kuo T-H. 2023. Minimal influence of estrous cycle on studies of female mouse behaviors. Front Mol Neurosci 16:1146109. doi:10.3389/fnmol.2023.1146109

Zhao W, Li Q, Ma Y, Wang Z, Fan B, Zhai X, Hu M, Wang Q, Zhang M, Zhang C, Qin Y, Sha S, Gan Z, Ye F, Xia Y, Zhang G, Yang L, Zou S, Xu Z, Xia S, Yu Y, Abdul M, Yang J-X, Cao J-L, Zhou F, Zhang H. 2021. Behaviors Related to Psychiatric Disorders and Pain Perception in C57BL/6J Mice During Different Phases of Estrous Cycle. Front Neurosci 15:650793. doi:10.3389/fnins.2021.650793